# Acceleration of the southern African easterly jet driven by radiative effect of biomass burning aerosols and its impact on transport during AEROCLO-sA

Jean-Pierre Chaboureau[1], Laurent Labbouz[1], Cyrille Flamant[2], and Alma Hodzic[3]

[1]Laboratoire d'Aérologie (LAERO), Université de Toulouse, CNRS, UPS, Toulouse, France
[2]LATMOS/IPSL, Sorbonne Université, CNRS, UVSQ, Paris, France
[3]National Center for Atmospheric Research, Boulder, CO, USA

**Correspondence:** Jean-Pierre Chaboureau (jean-pierre.chaboureau@aero.obs-mip.fr)

**Abstract.** The direct and semi-direct radiative effects of biomass burning aerosols (BBA) are investigated over southern Africa and the southeast Atlantic during the Aerosols, Radiation and Clouds in southern Africa (AEROCLO-sA) field campaign in September 2017. A reference convection-permitting simulation has been performed using the Meso-NH model with an on-line dust emission scheme, a strongly absorbing BBA tracer emitted using the daily Global Fire Emissions Database and online-computed backward Lagrangian trajectories. The simulation captures both the aerosol optical depth and the vertical distribution of aerosols as observed from airborne and space-borne lidars. The occurrence of stratocumulus over the southeast Atlantic, deep convective clouds over equatorial Africa and the large-scale circulation are all reproduced by the model. If the radiative effects of BBA are omitted in the model, we show that (i) the smoke plume is too low in altitude, (ii) the low-cloud cover is too weak, (iii) the deep convective activity is too frequent but not intense enough, (iv) the Benguela low-level jet is too strong, and (v) the southern African easterly jet is too weak. The Lagrangian analysis indicates that the radiative effect of BBA leads to the transport of BBA to higher altitudes, farther southwest and with a stronger diurnal oscillation. The higher smoke plume altitude can be explained by a combination of three factors: increased upward motion induced by the stronger southern African easterly jet, self-lofting of BBA and reduced subsidence associated with a less frequent deep convective activity over Western equatorial Africa.

## 1 Introduction

Biomass burning aerosols (BBA) are a major forcing agent of the regional climate over southern Africa and the southeast Atlantic during the burning season. From July to October, BBA are emitted by fires over the African continent and remain well-mixed in the boundary layer of 3 km deep on average over the source regions. They are transported by the prevailing mid-tropospheric easterly winds over the ocean, which is semi-permanently covered by an extensive stratocumulus cloud deck. Rich in black carbon that strongly absorbs solar radiation, BBA have a direct radiative effect (DRE) that changes sign depending on the underlying albedo i.e, BBA DRE is positive over highly reflective surfaces such as the stratocumulus deck and negative above low reflective surfaces such as the dark ocean. An accurate representation of the horizontal and vertical distribution of

aerosol and cloud properties is therefore essential for regional climate modeling. The large spread in the sign and magnitude of the DRE as predicted by climate models has motivated several field campaign studies over the southeast Atlantic (Zuidema et al., 2016).

BBA have also a semi-direct effect by affecting air temperature, atmospheric stability, low-level clouds and the regional atmospheric circulation. Tummon et al. (2010) showed a shallower boundary layer over the continent resulting from surface cooling combined with BBA-induced warming of the lower troposphere. Sakaeda et al. (2011) found an increased low cloud cover over the southeast Atlantic as a response to increased lower tropospheric stability and reduced large-scale subsidence. Das et al. (2020) showed that elevating the BBA layer to higher levels, in agreement with lidar observations, increases oceanic cloudiness near the coast south of 10° S and decreases it far from the coast. Hodzic and Duvel (2018) found found reduced deep convection over a tropical island and convergence of water vapor toward the island for moderately absorbing BBAs and increased deep convection for more strongly absorbing BBAs. Mallet et al. (2020) showed that vertical changes in air temperature limit the subsidence over the southeast Atlantic, creating a cyclonic anomaly in the lower atmosphere. Conversely, the increase in lower troposphere stability over Angola induces an anticyclonic anomaly. These changes in regional atmospheric circulation are crucial for the path of rivers of smoke, from the BBA sources in the tropics to their transport to south America (Holanda et al., 2020), temperate mid latitudes and the southwestern Indian Ocean (Flamant et al., 2022).

A dynamical aspect left unmentioned is the Southern African Easterly Jet (AEJ-S), a thermal wind resulting from the temperature gradient between the semi-arid region of southern Africa and the Congo Basin. Kuete et al. (2020) showed that the AEJ-S is maintained by a mid-level high that forms over the Kalahari region during September to November. Adebiyi and Zuidema (2016) analyzed BBA under strong and weak AEJ-S conditions for a 10 y period. They found that the AEJ-S speed maximum during September and October coincides with the maximum aerosol optical depth over the southern Africa and the maximum low cloud fraction over the southeast Atlantic. They also showed that the AEJ-S transports BBA more efficiently over the southeast Atlantic during strong jet episodes than during weak jet episodes. To our knowledge, the radiative effects of the BBA on the AEJ-S has not been studied to date, whereas the importance of the radiative effects of another absorbing aerosol (dust) on the northerly AEJ over West Africa has been demonstrated by Tompkins et al. (2005) and Lavaysse et al. (2011) among others.

The objective of this study is to examine the direct and semi-direct radiative effects of BBA over southern Africa and the southeast Atlantic with a particular focus on the AEJ-S. To achieve this objective, we investigate their effects using the airborne assets deployed during the AErosol, RadiatiOn, and CLOuds in southern Africa (AEROCLO-sA) field campaign (Formenti et al., 2019). From 5 to 12 September 2017, airborne lidar and dropsonde observations provided dedicated measurements of atmospheric dynamics, thermodynamics and aerosol composition. We run two ensembles of convection-permitting simulations during 16 d, one with the radiative effects of BBA, the other without. Running the model without deep convection parameterization allows us to explicitly represent the vertical motions in deep convection and to calculate the trajectory of the air parcels in the updrafts and downdrafts. We compare both ensembles with satellite and ground-based observations, as well as airborne lidar and dropsonde measurements. After showing that the overall realism of the ensemble with radiatively-active BBA on aerosol and cloud distribution is significantly improved, we discuss the BBA radiative effects in accelerating the AEJ-S and

in modifying the horizontal and vertical transport of BBA in comparison with the ensemble of radiatively-inactive BBA simulations. Note that September 2017 has a slightly weaker AEJ-S than the climatological mean (Ryoo et al., 2021) and that the microphysical BBA-cloud interaction (the indirect effect of BBA) is not considered here.

The paper is organized as follows: Sect. 2 describes the data and the methods. Section 3 gives an overview of aerosols, clouds and dynamics during the 16 d period of interest. It also evaluates the simulations against observations and shows the superiority of the simulation with radiatively-active BBA. Section 4 analyzes the radiative impact of BBA on radiation, temperature, dynamics and transport. Section 5 concludes the paper.

## 2 Data and methods

### 2.1 Meso-NH convection-permitting simulations

The two ensembles of simulations are run with the nonhydrostatic mesoscale model Meso-NH (Lac et al., 2018), version 5.4, over a domain covering southern Africa and the southeast Atlantic (Fig. 1). The model grid has a 12 km horizontal spacing and 67 levels with a resolution of 60 m close to the surface to 600 m at high altitude. Thermodynamic and other scalar variables are advected with the piece-wise parabolic method (PPM) while the momentum variables are advected with a fourth-order centered scheme coupled to an explicit fourth-order centered Runge-Kutta time splitting (Lunet et al., 2017).

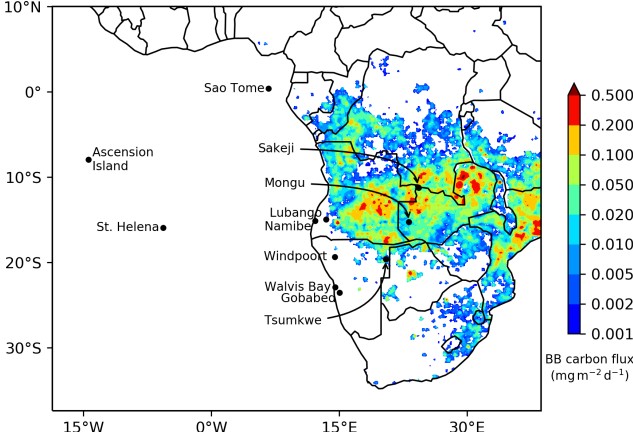

**Figure 1.** Meso-NH domain. The color shading shows the GFED emission of biomass burning carbon averaged between 1 and 16 September 2017.

The simulations use the Surface Externalisée (SURFEX) scheme for surface fluxes (Masson et al., 2013), a 1.5-order closure scheme for turbulence (Cuxart et al., 2000), an eddy-diffusivity mass-flux scheme for shallow convection (Pergaud et al., 2009), the Rapid Radiative Transfer Model (Mlawer et al., 1997) for longwave radiation and the two-stream scheme (Fouquart and Bonnel, 1986) for shortwave radiation. The cloud model is the bulk microphysical scheme for mixed-phase clouds of Pinty and Jabouille (1998). In addition, a subgrid-scale scheme for cloud cover and condensate is necessary when running the model at

12 km. We use the statistical scheme of Chaboureau and Bechtold (2002) wherein the cloud cover and the cloud condensate are function of the grid-scale saturation deficit. We however do not use any deep convection parameterization in order to allow an explicit convective transport of the smoky air parcels. The aerosol model has two components. The dust scheme of Grini et al. (2006) includes the Dust Entrainment and Deposition (DEAD) scheme (Zender et al., 2003) and the ORganic and Inorganic Log-normal Aerosols Model (ORILAM) (Tulet et al., 2005), which describes the transport, dry, and wet deposition of dust. The BBA scheme includes a biomass burning carbon tracer that is emitted from the daily Global Fire Emissions Database (GFED) version 4, available at a spatial resolution of $0.25°$ (van der Werf et al., 2017). The BB tracer is emitted in the first vertical layer of the model, allowing its mixing by turbulence within the boundary layer. No more elaborate injection assumptions or BBA removal processes are considered here because they were found in a multimodel evaluation of BBA plume transport to play a minor role (Das et al., 2017). The two ensembles of simulations differ by the radiative effect of BBA. NORAD assumes no radiative effect while BBRAD considers a mass extinction efficiency of $5.05\,\mathrm{m^2\,g^{-1}}$ at 532 nm for BBA as used by Mallet et al. (2020) for aged smoke. The single scattering albedo (SSA) is equal to 0.85. This value, which corresponds to strongly absorbing aerosols, is close to the vertical average estimated at Ascension Island (Wu et al., 2020) and over the southeastern Atlantic (Pistone et al., 2019; Cochrane et al., 2022).

The BBA transport is analyzed using the Lagrangian trajectories computed online in the model with three passive tracers (Gheusi and Stein, 2002). At each grid point of the simulation domain, the tracers are initialized with the 3-D field of their initial coordinates and advected by PPM, which conserve the mass properties of the tracers with low numerical diffusion. This allows to follow the 3-D position of each BBA tracer. Note that this approach takes into account any diabatic processes including BBA radiative heating allowing any cross-isentropic trajectory to be followed. To assess the convective activity in the simulations, synthetic brightness temperatures (BTs) at $10.8\,\mu\mathrm{m}$ are computed from the model three-hourly outputs using the radiative transfer model for the TIROS Operational Vertical Sounder (RTTOV) code (Saunders et al., 2018), implemented in Meso-NH by Chaboureau et al. (2008).

To assess the significance of changes due to the radiative effect of BBA, we follow the methodology of Das et al. (2020) and performed an ensemble of five members for each of the two radiative configurations. The members are generated by shifting their initial time by 6 h. The first member starts at 00:00 UTC 01 September 2017, the second one at 06:00 UTC, the third one at 12:00 UTC, the fourth at 18:00 UTC and the last one at 00:00 UTC 02 September 2017. The five members are then integrated until 00:00 UTC 17 September 2017. The initial and lateral boundary conditions are given by operational analyses of the European Center for Medium-Range Weather Forecasts (ECMWF). In the following, the ensemble means of BBRAD and NORAD are referred to as the 5-member ensemble and statistical significance at the 0.05 level to the two-tailed Student's $t$ test. The 7 first day is used to spin the model up and the results are shown for averages between 8 and 16 September. It corresponds to the time required for the BBAs to reach the westernmost islands in the Atlantic, as shown below. It also allows the BBAs to have a full radiative impact on the atmospheric circulation while keeping them consistent with temperature and the winds.

## 2.2 Observations

Lidar LEANDRE (Lidar Embarqué pour l'Etude des Aérosols, Nuages, Dynamique, Rayonnement et Espèces minoritaires) Nouvelle Génération (LNG, Bruneau et al., 2015) and dropsonde observations were acquired from the SAFIRE (Service des Avions Français Instrumentés pour la Recherche en Environnement) Falcon 20. The aircraft was based in Walvis Bay, located on the west coast of Namibia. Here, we use data acquired on 6 September 2017 between 07:18 and 09:12 UTC during which the Falcon 20 flew over continental Namibia prior to flying over the ocean (Formenti et al., 2019). Extinction at 532 nm is retrieved using a standard lidar inversion method that employs a lidar ratio of 70 sr, characteristic of BBA in the free troposphere (a detailed description of the inversion method is given in Chazette et al., 2019). The retrievals have an estimated uncertainty of 15 %, and a resolution of 2 km in the horizontal and 15 m in the vertical. Profiles from Vaisala dropsondes released at 08:43 and 09:08 UTC are also used.

Over the remote areas of the AEROCLO-sA domain, the vertical distribution of aerosols is assessed using the CATS (Cloud-Aerosol Transport System) lidar operated from the International Space Station (Yorks et al., 2016). CATS has flown over the simulation domain several times. We select two orbits, one over Angola and Congo on 14 September between 19:02 and 19:11 UTC, the other over the southeast Atlantic on 8 September between 22:40 and 22:47 UTC. We use the CATS Level 2 Layer extinction at 1064 nm available at 60 m vertical and 5 km horizontal resolution.

Aerosol optical depth (AOD) is evaluated using the daily merged Level 3 AOD product (Deep Blue and Dark Target) from the MODIS 6.1 collection available at a spatial resolution of 1°. This product provides broad coverage by combining clear land surface (Deep Blue) retrievals with ocean and vegetated land surface (Dark Target) retrievals. We select the Aqua platform retrievals because it crosses the equator at 13:30 LT when the boundary layer is well developed. The AOD evaluation also takes advantage of the Aerosol Robotic Network (AERONET) sun photometers (Holben et al., 1998) at 10 selected stations available on the simulation domain (Ascension Island, Gobabeb, Lubango, Mongu, Namibe, St. Helena, Sakeji, São Tomé, Tsumkwe and Windpoort; Fig. 1). The sun photometers provide the aerosol properties during the daytime. Here, we use the daily-mean, cloud cleared and quality assured, level 2.0 product of the AOD at 532 nm, the central wavelength of the solar window, and the SSA at 440 nm, the closest available wavelength to 532 nm.

Deep convective clouds are assessed using the BTs at 10.8 µm obtained with the Spinning Enhanced Visible and InfraRed Imager (SEVIRI) instrument aboard the geostationary satellite Meteosat Second Generation (MSG). The BTs are re-gridded from the 4 km resolution (at nadir) to the horizontal resolution of the simulations of 12 km. Following Söhne et al. (2008) among many others, deep convective clouds are defined as grid points with BTs less than 230 K. Low-level clouds are evaluated using level 3 daily product of cloud fraction and cloud top temperature from the MODIS collection 6.1 available in 1° spatial resolution. For consistency with the MODIS AOD product, we use these retrievals from the Aqua platform. The low-level cloud fraction is defined as the cloud fraction for which the cloud top temperature is greater than 273 K.

## 3    Overview of aerosols, clouds and dynamics

### 3.1    Horizontal distribution

The time variation in AOD is examined at 10 AERONET stations available on the domain and for the BBRAD and NORAD members starting at 00:00 UTC 01 September 2017 (Fig. 2). The SSA is also shown for AERONET and BBRAD with dotted lines. In order to produce a column SSA value for comparison with AERONET, the BBRAD simulated SSA is averaged after weighting its value according to the aerosol optical depth at each vertical level. In the simulations, AOD is due to BBA and dust, with the contribution of the latter shown by thin lines. Regardless of the station, the model lacks AOD at the beginning of the simulations. This is due to the time required for aerosols to be transported from the sources to the stations. Since the aerosol sources in the model are either BB or dust, both located in southern Africa, a few days are needed for the aerosols to reach the model boundaries. Note that AOD due to dust mainly affects Gobabeb, and only on 3 and 4 September. Dust has hence a negligible contribution to the simulated AOD most of the time and for most stations.

The long spin-up of BBA explains the zero AOD values simulated at Ascension Island and St. Helena stations in the first 6 days (Fig. 2a and c). The AOD at these stations varies strongly with the simulated circulation. On the one hand, BBRAD reproduces the maximum of 0.5 on 10 September at St. Helena while it is absent in NORAD (Fig. 2c). On the other hand, NORAD overestimates the AOD at São Tomé while BBRAD simulates it in the right range (i.e., around 0.2) in the last 10 days of the simulation (Fig. 2e). In NORAD, the BBA plume extends too far north and not far enough west. At Sakeji and Mongu, AOD is observed between 0.5 and 1 (Fig. 2b and d). Both simulations show the correct range of AOD. BBRAD slightly overestimates the minimum SSA value of 0.85 observed at Mongu while capturing its variation around 0.87 at Sakeji. This suggests that the regular emission of BBA is well simulated as well as its optical properties. Near the fire source areas, Lubango, Namibe and Windpoort show a stronger variation with AOD between 0.3 and 1.3 (Fig. 2f–h). The simulations reproduce this variation well, but underestimate the AOD peak by a few tenths. BBRAD reproduces the AERONET SSA value around 0.86 very well. For other stations further south over Namibia (Fig. 2i and j), AOD shows less variability, around 0.5 at Gobabed and Tsumkwe. These features are fairly well reproduced by both simulations, including the short period when dust contributes to the AOD. The observed SSA value between 0.85 and 0.90 is also well simulated by BBRAD.

The simulations are further assessed in terms of horizontal distribution of AOD, low-level and deep convective clouds (Fig. 3). The fields are ensemble means and averaged between 8 and 16 September 2017 because of the long spin-up of BBA tracer as shown previously. They are daily averages for deep convective clouds taking advantage of frequent observations by the geostationary SEVIRI instrument and averages at 12:00 UTC for AOD and low-level clouds, a time close to the observations done by MODIS aboard the Aqua satellite. The dynamics is represented by the wind field averaged at 12:00 UTC and taken at different altitudes: 12 km corresponds to the level of maximum detrainment by deep convection over equatorial Africa; 4 km is the altitude where the AEJ-S is maximum and rapidly transports BBA; 1 km above ground level (a.g.l.) represents the low-level circulation which has an impact on the occurrence of low-level clouds.

Deep convective clouds are observed mostly over the Congo Basin and north of the equator (Fig. 3a). A few events are embedded in the mid-latitude circulation south of 30° S. The simulations reproduce well the occurrence of deep convective clouds

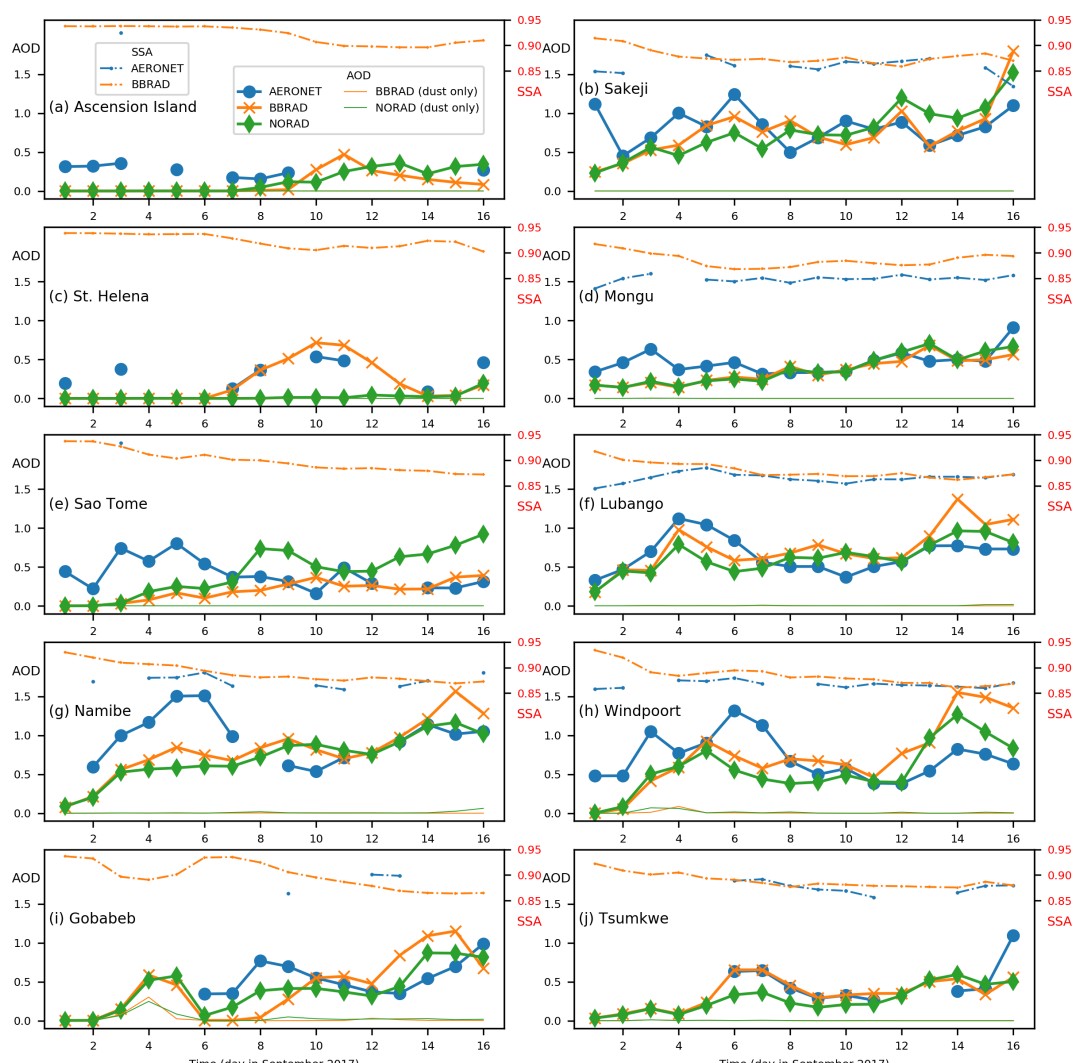

**Figure 2.** Time evolution of daily mean AOD at 532 nm between 1 and 16 September 2017 from AERONET (blue), BBRAD (orange) and NORAD (green) at **(a)** Ascension Island, **(b)** Sakeji, **(c)** St. Helena, **(d)** Mongu, **(e)** São Tomé, **(f)** Lubango, **(g)** Namibe, **(h)** Windpoort, **(i)** Gobabeb and **(j)** Tsumkwe. The orange and green thin lines show the AOD due to dust for BBRAD and NORAD simulations, respectively. The blue and orange dotted lines show the SSA at 440 nm for AERONET and BBRAD, respectively. Results are shown for the BBRAD and NORAD members starting at 00:00 UTC 01 September 2017.

as well as the areas free of deep convection, i.e. most of the southern Africa and the southeastern Atlantic. In the observations, the occurrence of deep convective clouds equals 3.0 % over land and 7.7 % over the yellow box shown in Fig. 3a–c. NORAD is more active in deep convective clouds with 3.1 and 8.2 % of occurrence versus 2.8 and 7.7 % for BBRAD, respectively. This change in deep convective clouds with radiatively active BBA is a semi-direct effect discussed in Sect. 4.3. The wind at 12 km shows an anticyclonic circulation with strong westerlies over southern Africa and easterlies above $-12\,\mathrm{m\,s^{-1}}$ at the equator,

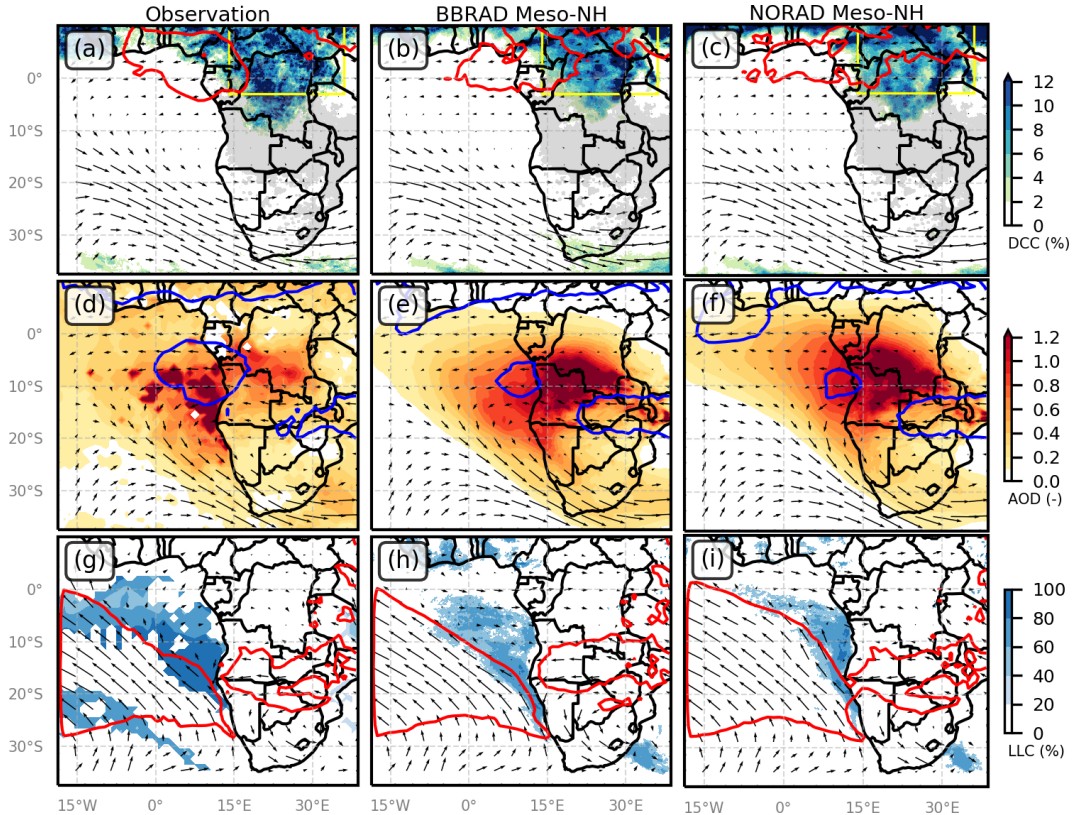

**Figure 3.** Top panel: occurrence of deep convective clouds from **(a)** SEVIRI, **(b)** BBRAD and **(c)** NORAD. Fire sources are shaded in grey. Middle panel: AOD from **(d)** MODIS, **(e)** BBRAD and **(f)** NORAD. Bottom panel: occurrence of low-level clouds from **(g)** MODIS, **(h)** BBRAD and **(i)** NORAD. Arrows show the winds at **(a–c)** 12 km, **(d–f)** 4 km and **(g–i)** 1 km a.g.l. from **(a, d, g)** ECMWF, **(b, e, h)** BBRAD and **(c, f, i)** NORAD. Contours show the zonal wind at **(a–c)** $-12$, **(d–f)** $-8$ and **(g–i)** $-4\,\mathrm{m\,s^{-1}}$ at 12, 4 and 1 km, respectively. Fields are averaged between 8 and 16 September 2017, and BBRAD and NORAD fields are ensemble means.

along the southern fringe of the tropical easterly jet, for all three datasets. The tropical easterly jet is stronger for NORAD than for the ECMWF analysis and for BBRAD in coherence with the more active deep convection feeding it.

AOD values retrieved from MODIS observation show an arc of high values spanning over the Atlantic ocean, between 10 and 20° S and along the border between Angola and the Democratic Republic of the Congo (Fig. 3d). BBRAD reproduces this feature well, but with higher values along the border and lower values over the ocean. NORAD mimics BBRAD along the border, but misses the area over the ocean. Instead, it spans large values of AOD along the African coast up to the Gulf of Guinea. As for the wind at 12 km, the wind at 4 km shows an anticyclonic circulation with strong westerlies over southern Africa and easterlies above $-8\,\mathrm{m\,s^{-1}}$ around 10° N. The value of $-8\,\mathrm{m\,s^{-1}}$ is chosen because this threshold provides a good identification of the S-AEJ. Compared to the ECMWF analysis, NORAD overestimates the latter more than BBRAD. The other

branch of strong easterlies around 15° S over land and 8° S over sea is the AEJ-S. BBRAD accelerates the AEJ-S compared to

NORAD. It however overestimates the AEJ-S over Namibia and Tanzania with respect to ECMWF.

Low-level clouds are observed mainly over the southern Atlantic with cloud fraction up to 80 % off Namibia (Fig. 3g). Another band lies south of 20° S over the open ocean which is not simulated. Both ensembles of simulations reproduce the low-level cloud cover off Namibia, but over too small an area and with too low a cloud fraction for NORAD. BBRAD mimics the observed low-level cloud cover as well as the analyzed wind field, in particular the Benguela low-level jet off Namibia.

Extension too far north of the low-level jet for NORAD suggests that the dynamics inhibit the low-level cloud fraction at this location.

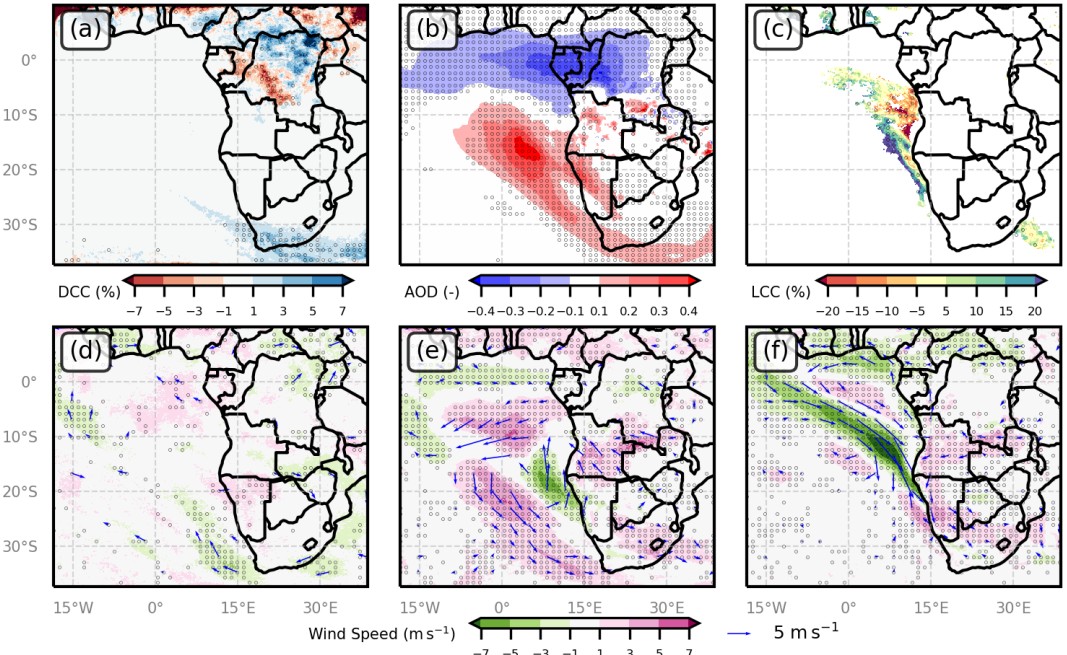

**Figure 4.** Changes between BBRAD and NORAD ensemble means in **(a)** occurrence of deep convective clouds, **(b)** AOD, **(c)** occurrence of low-level clouds, and wind speed at **(d)** 12 km, **(e)** 4 km and **(f)** 1 km a.g.l. Black dots indicate where changes in deep convective clouds, AOD, low-level clouds and wind speed are statistically significant at the 0.05 level. Arrows indicate wind field anomalies when significant at the 0.05 level. Fields are averaged between 8 and 16 September 2017.

The changes between BBRAD and NORAD and their significance at the 0.05 level following the $t$ test are shown in Fig. 4. Two significant radiative effects of BBA on the occurrence of deep convection clouds are (i) to move them over the Democratic Republic of the Congo closer to the southwest and (ii) increase them over southern South Africa (Fig. 4a). This is associated

with a slight increase in wind speed at 12 km altitude over the Gulf of Guinea and a decrease over Ascension Island and off the southwestern tip of southern Africa (Fig. 4d). A strong significant impact is the change in AOD that occurs outside the fire source areas (Fig. 4b). The activation of the radiative properties of BBA decreases the AOD by a few tenth over the equator and

increases it over the South Atlantic, west of the fire source areas and at the southern end of Africa. In other words, the BBA are exported from their source areas more to the south when they are absorbing than when they are not. This enhanced export

to the south is explained by the strong significant change in wind speed at 4 km altitude (Fig. 4e). First, the AEJ-S over Angola and its oceanic branch along $10°$ S is accelerated by a few m s$^{-1}$. The increase in easterly winds strengthens the anticyclonic circulation, which leads to an acceleration of the westerly winds over the South Atlantic. Second, the wind speed decreases over the Gulf of Guinea, which limits the export of BBA in this region. This decrease in wind speed occurs also at 1 km a.g.l (Fig. 4e). The slowing of the Benguela low-level jet and its extension poleward are more spectacular. This contributes to the

greater presence of low-level clouds along the southwestern boundary of the stratocumulus deck (Fig. 4c).

## 3.2 Vertical distribution

Previous evaluations of BBA simulations revealed a systematic smoke layer 1–2 km too low compared to lidar observations (Das et al., 2017, 2020). Here, we evaluate the vertical distribution of extinction using two regional cross-sections obtained by CATS and a mesoscale measured by LNG. This evaluation is done for the BBRAD and NORAD members starting at

215 00:00 UTC 01 September 2017. Two examples of the vertical distribution of extinction at 1064 nm observed by CATS are shown, one over the fire source regions on 14 September, the other over the stratocumulus region on 8 September (Fig. 5). They are compared to the extinction due to BBA and dust and to the cloud fraction from the simulations.

On 14 September, a well-mixed BBA layer with extinction value above 0.05 km$^{-1}$ and a depolarization ratio less than 0.1 (not shown) is present between the ground and 5 km altitude (Fig. 5d). Above Angola, i.e. between 10 and $15°$ S, the layer is

220 shallower and extinction is often constant in columns between the ground and 4 km altitude with values up to 0.25 km$^{-1}$. Over the Namibian desert, i.e. around $18°$ S, the depth of the BBA layer increases and the extinction values decrease with altitude. Over ocean, the BBA layer is over the planetary boundary layer up to $27°$ S and 6 km altitude where it is topped by clouds with large extinction values. In the first 2 km altitude, aerosols are present below shallow clouds. Note that along the coast around $21°$ S, the strong extinction value in the first km is associated with a depolarization ratio greater than 0.5 (not shown) suggesting

the presence of dust. BBRAD mimics both vertical and horizontal extinction reasonably well (Fig. 5e). Above Angola, the BBA layer is well-mixed below the approximately 316 K isentrope located at 4 km altitude. As for the observations, extinction can reach values as high as 0.25 km$^{-1}$. Over Namibia, two layers are simulated, one under the 316 K isentrope and another between 320 and 324 K, that is up to 5 km altitude as observed. BBA successfully simulates the decoupling of the layer over the ocean due to the tilting of the isentropes with latitude. In the first 2 km, BBRAD well simulates a dusty cloud plume near the coast.

Away from the coast, only the shallow clouds are simulated suggesting that the observed aerosol signal is neither due to BBA nor to dust, but rather to sea salt. NORAD does simulate a BBA layer sandwiched between the 304 and 320 K isentropes (Fig. 5f). However, its depth over Angola is overestimated by about 1 km and its southern extent over the ocean is limited to $24°$ S. Counter-intuitively, the heat low over Namibia is warmer and deeper for NORAD than for BBRAD. All these features can be explained by the stronger AEJ-S in BBRAD. Indeed, a stronger jet limits the vertical development of the boundary

layer over Angola. It also strengthens the cyclonic circulation that weakens the heat low and exports the BBA further west,

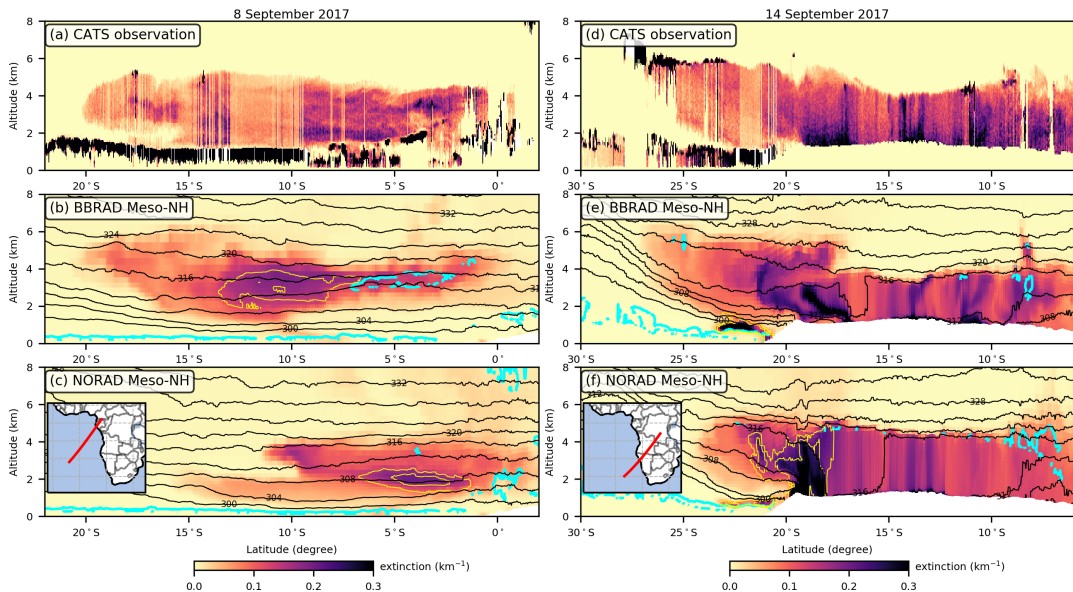

**Figure 5.** Vertical cross-sections of extinction at 1064 nm on **(a–c)** 8 and **(d–f)** 14 September 2017 from **(a, d)** CATS, **(b, e)** BBRAD and **(c, f)** NORAD along the red line shown in the inset of the bottom panels. In panels **(b, c, e, f)**, the black contours show the potential temperature (in K), the cyan contour the cloud fraction at 10 %, and the yellow contours the dust extinction at 0.05 and 0.1 km$^{-1}$. On 8 September, CATS observations were taken between 22:40 and 22:47 UTC and Meso-NH simulations are at 24:00 UTC. On 14 September, CATS observations were taken between 19:02 and 19:11 UTC and Meso-NH simulations are at 24:00 UTC. Results are shown for the BBRAD and NORAD members starting at 00:00 UTC 01 September 2017.

thereby limiting their direct radiative effect over land. NORAD correctly simulates the dusty cloud plume near the coast, but incorrectly fills the boundary layer between 18 and 20° S with mainly dust.

On 8 September, low-level clouds are ubiquitous along the track (Fig. 5a). Their thickness is about 500 m and their base varies from 1 km on the southern part of the track to around 500 m in its center. Between 5 and 15° S, their tops touch the BBA
layer which extends up to 5 km of altitude. The BBA layer is characterized by multiple superimposed layers of few-hundred-meter thick with different extinction values (and a low depolarization ratio; not shown). The largest extinction values are north of 8° S and south of 13° S. The lower values in between suggest that BBA circulate around this area. Between 16 and 20° S, the BBA layer is 1 km above the boundary layer. BBRAD shows a BBA structure between 0 and 20° S and 1 and 6 km altitude, which is close to what is observed (Fig. 5b). The layer with high values of extinction between 5 and 10° S is however thinner
than observed and the stratocumulus deck is too shallow. Interestingly, dust contributes a little to the extinction at 3 km around 8–13° S with values around 0.05 km$^{-1}$. NORAD misses the occurrence of elevated BBA between 15 and 20° S (Fig. 5c). Instead, it transports BBA up to 4 km at 12° S. The stratocumulus deck is too shallow and less continuous than observed. It incorrectly simulates a heavy dusty layer between 2 and 8° S.

Another difference in the vertical distribution of aerosols between the simulations is shown for the BBA layer observed by LNG off Namibia on 6 September and on the vertical profiles of potential temperature and wind speed (Fig. 6). Note that no dust is found along the leg for both simulations. The BBA layer observed by LNG is a well-mixed layer between 3 and 5.5–6 km altitude (Fig. 6a). It is partially topped by shallow clouds as shown by the large extinction values. Dropsondes at 08:43 and 09:08 UTC show that the potential temperature is almost constant with altitude, around 316 K (Fig. 6d, e). The same is true for the wind speed, around $14\,\mathrm{m\,s^{-1}}$ at 08:43 UTC and $17\,\mathrm{m\,s^{-1}}$ at 09:08 UTC (Fig. 6f, g). Over the ocean, the BBA layer is thinner and stratocumulus are found near the surface.

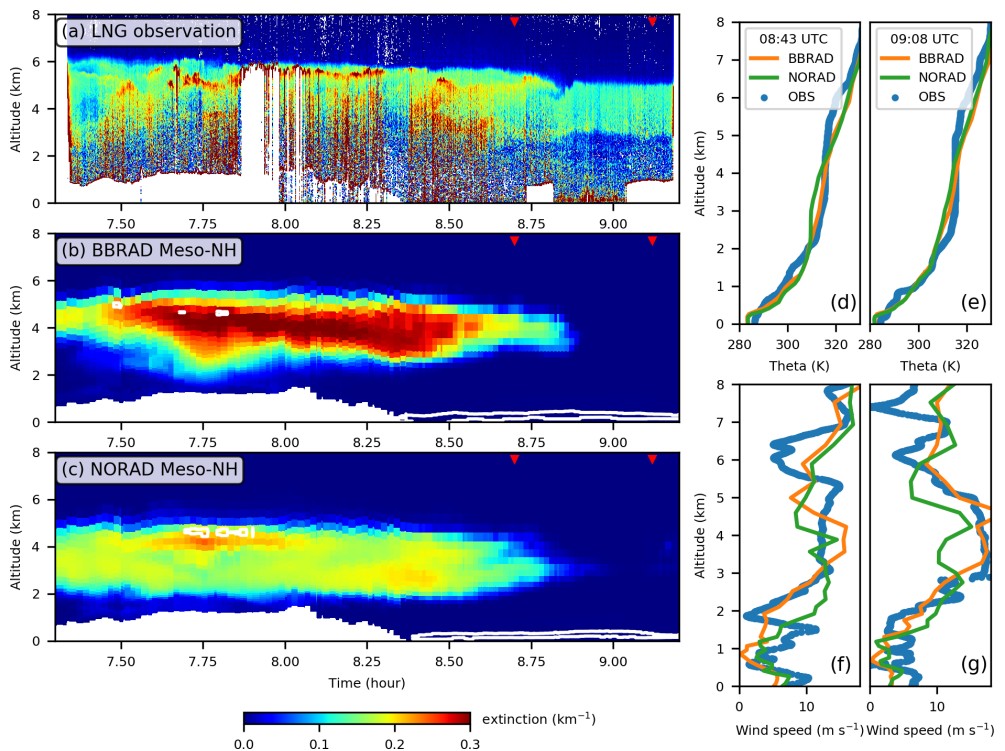

**Figure 6.** Left column: vertical cross-sections of extinction at 532 nm on 6 September 2017 from **(a)** LNG, **(b)** BBRAD and **(c)** NO-RAD along the F20 track shown in Fig. 7. LNG observations were taken between 07:18 and 09:12 UTC and Meso-NH simulations are at 09:00 UTC. the white contours show the cloud fraction at 10 %. The red triangles locate the dropsondes. Right column: profiles of **(d, e)** potential temperature and **(f, g)** wind speed from dropsondes released at **(d, f)** 08:43 and **(e, g)** 09:08 UTC, BBRAD and NORAD at 09:00 UTC. Results are shown for the BBRAD and NORAD members starting at 00:00 UTC 01 September 2017.

The simulations reproduce the stratocumulus above the ocean and the BBA layer over land quite well (Fig. 6b and c). However, the thinner layer observed over the ocean (after 08:45 UTC, i.e. 8.75 h) is missing. The BBA top is also not high enough by 500 m. Therefore, the correctly but too few simulated clouds at the top of the BBA layer are too low by 500 m. This is consistent with the inversion of the potential temperature being too low by 500 m. Between 3–4 km altitude, the temperature and the wind speed are higher for BBRAD than for NORAD and closer to the observations. This is expected from the radiative

effect of BBA on temperature and AEJ-S. BBRAD simulates a thinner BBA layer, which also agrees with the observations. It produces a much larger extinction than NORAD, which is somewhat overestimated.

To investigate the difference in extinction between the simulations, a sample of randomly selected back-trajectories of smoky air parcels reaching the LNG track are shown in Fig. 7. For both simulations, most of them stretch zonally between Zambia and

Angola at an altitude of 3–4 km, i.e. within the AEJ-S. At the jet entrance, the area of convergence of the jet inflow is located over northern Botswana and Zimbabwe, which explains why a large part of the trajectories originate there. Another feature of interest shared by both simulations is the area of divergence of the AEJ-S in the Angola coastal region. They both show BBA at 4–5 km altitude. Indeed, the ageostrophic circulation in the jet outflow as shown by Adebiyi and Zuidema (2016) is expected to favor the passage of smoky air parcels below or above the jet level. This explains why many trajectories go up there.

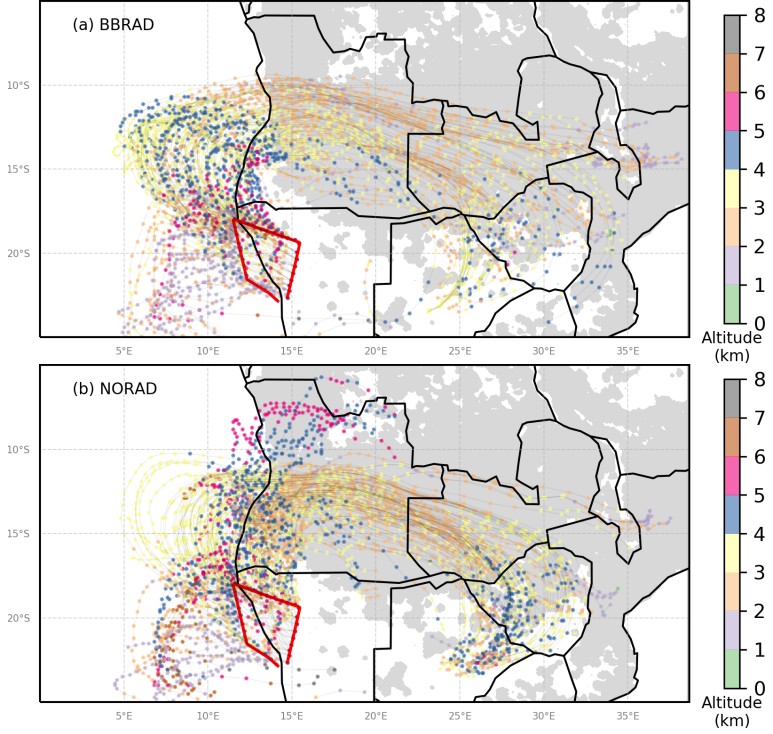

**Figure 7.** Selected trajectories of BB tracers ending along the LNG track (red line) at 09:00 UTC on 6 September 2017 in **(a)** BBRAD and **(b)** NORAD. Their position is indicated by dots every 3 h. Their length is 5 d and 9 h. Fire sources are shaded in grey color. Results are shown for the BBRAD and NORAD members starting at 00:00 UTC 01 September 2017.

In NORAD only, the BBA that are located at 5-6 km altitude in northern Angola and then transported southward are under the dynamical forcing of the AEJ-S. In BBRAD, the track of the trajectories is narrower with more trajectories coming from the north over the continent and some going further west over the ocean. As a result, air masses with partly oceanic trajectories travel longer between the fire region and the LNG track than air masses with shorter, predominantly continental trajectories. Since the northern fire sources are more active, smoky air parcels with larger extinction values are simulated along the LNG

track. In other words, a stronger AEJ-S leads to a longer pathway over fires (i.e., BBA sources) and higher extinction values in this specific case.

## 4   Impacts of BBA on the atmosphere

### 4.1   Direct effect on radiation

The direct effect of BBA on shortwave (SW) radiation and radiative heating is examined for BBRAD between 8 and 16 Septem-
ber (Fig. 8). It is shown as a daily mean to be discussed with results of climate models reported in the literature. The DRE is calculated as the difference between the total SW flux and the SW flux without the contribution of the BBA.

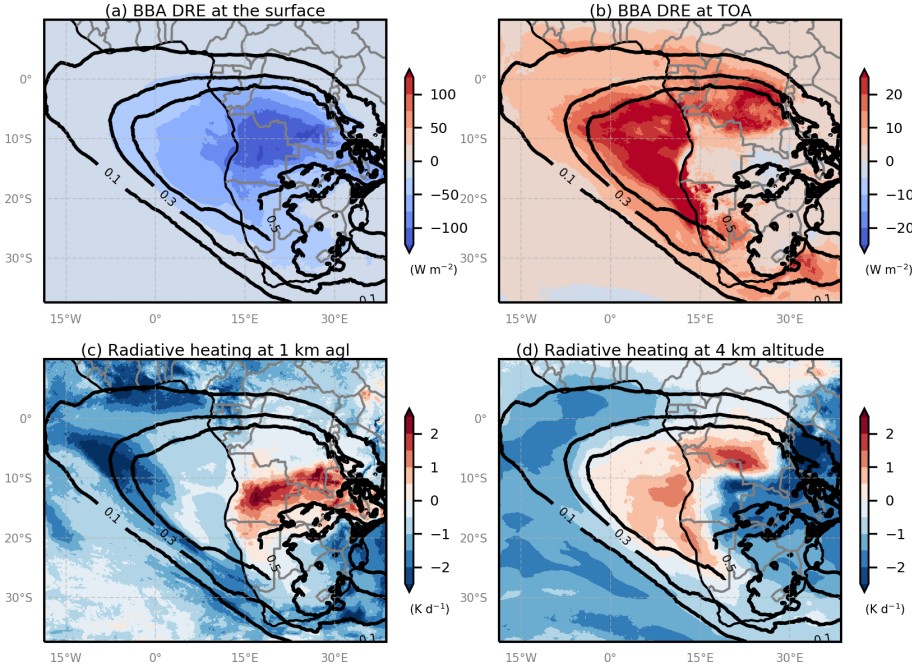

**Figure 8.** Results from the BBRAD ensemble mean: BBA DRE (W m$^{-2}$) at **(a)** the surface and **(b)** top of the atmosphere (TOA), and radiative heating (K d$^{-1}$) at **(c)** 1 km a.g.l. and **(d)** 4 km altitude. Fields are averaged daily between 8 and 16 September 2017.

At the surface, the DRE is negative or zero everywhere (Fig. 8a). It varies with AOD, with a dimming effect of BBA as strong as $-100$ W m$^{-2}$ over and around Angola. It is 2–3 times stronger than the seasonally averaged values of $-30$ W m$^{-2}$ reported by Sakaeda et al. (2011) and Mallet et al. (2020). The larger magnitude is explained by both the greater insolation and
AOD in mid-September compared to seasonal values. In terms of radiative forcing efficiency, i.e. DRE divided by AOD, it is between $-10$ and $-20$ W m$^{-2}$ $\tau^{-1}$, a value consistent with the estimate of Sakaeda et al. (2011). At the top of the atmosphere, the DRE is mainly positive (Fig. 8b). It is strongest over ocean above the stratocumulus deck where it reaches a maximum of 40 W m$^{-2}$ while it is around zero over Angola and Zambia and a small part of the Gulf of Guinea. This uneven DRE over

areas of high AOD is due to the surface albedo, which is high over the oceanic stratocumulus deck and low over the land and stratocumulus-free parts of the ocean. Again, these values are 2–3 times stronger than the seasonal mean values by Sakaeda et al. (2011) and Mallet et al. (2020). Furthermore, the DRE is expected to be negative over low reflectance, high AOD surfaces, i.e. over Angola and the Congo Basin, whereas near zero values are obtained here.

At 1 km a.g.l., the radiative heating is positive over the fire source areas (Fig. 8c). It reaches a maximum of $2\,\mathrm{K\,d^{-1}}$ over central Angola and southeastern Democratic Republic of the Congo. Elsewhere, the radiative heating is negative. This is mainly a night-time cooling effect (due to water vapor), which is the strongest over the cloud-free equatorial ocean. At 4 km altitude, the radiative heating is positive almost everywhere where the AOD is greater than 0.5, with maximum values around $1\,\mathrm{K\,d^{-1}}$ (Fig. 8d). Elsewhere, the radiative heating is negative. Over Zambia and Tanzania, the cooling that occurs just above the BBA layer is one of the strongest. Overall, the heating rates due to BBA are within the range of daily values around $1\,\mathrm{K\,d^{-1}}$ reported by Keil and Haywood (2003), Tummon et al. (2010), and Mallet et al. (2020), among others.

## 4.2 Impact on dynamics

A vertical perspective along latitude is shown over ocean, across the stratocumulus deck between 5 and 10° E, and over land, across the BBA source areas between 15 and 25° E (Fig. 9). The latitude-height cross-section is used to show the S-AEJ thermal wind for which the zonal wind varies with the meridional temperature gradient. Fields are averaged at 12:00 UTC between 8 and 16 September. Examination at noon time allows a highlighting of the radiative heating in the BBA layers while excluding nighttime cooling. Changes between BBRAD and NORAD and their significance are shown in Appendix A (Figs. A1 and A2.)

Over land, the radiative heating at the BBA layer reaches a maximum value of $12\,\mathrm{K\,d^{-1}}$ at 2 km altitude around 12° S (Fig. 9b). It shows values higher than $2\,\mathrm{K\,d^{-1}}$ up to 3–4 km altitude above the fire source region between 8–22° S. This fits with the atmosphere slice with extinction higher than $0.1\,\mathrm{km^{-1}}$ in BBRAD (Fig. 9d). This strong heating warms the well-developed boundary-layer, lowering the 316 K isentrope to less than 4 km in altitude between 10–20° S and reinforcing the meridional contrast in potential temperature. Due to the thermal wind balance, the AEJ-S shows a core value of $-10\,\mathrm{m\,s^{-1}}$ at 12° S while the associated ageostrophic circulation occurring in the entrance region of the AEJ-S induces upward motion to its north and downward motion to its south (Adebiyi and Zuidema, 2016). Self-lofting, i.e. the increase in buoyancy in the BBA layer due to radiative heating, also enhances upward motion. Because of the strong AEJ-S, the extinction is less developed vertically between 10–20° S while it has an ascending branch at about 8° S. In NORAD, the core value of the AEJ-S equals $-8\,\mathrm{m\,s^{-1}}$ and the vertical motion is mainly upward in the first 8 km. The largest upward motions are in the Hadley cell north of 0° and in the boundary layer over the African highlands. It results in an extinction field almost evenly distributed vertically below the 316 K isentrope. In summary, accounting for the radiative effect of BBA changes the dynamics that feedback on the vertical distribution of BBA extinction, elevating them north of the AEJ-S.

Over ocean, the radiative heating within the BBA layer reaches a maximum value of $10\,\mathrm{K\,d^{-1}}$ at 4 km at 12° S (Fig. 9a). This value fits the SW heating rate of $9\,\mathrm{K\,d^{-1}}$ inferred from satellite observations (Deaconu et al., 2019). Located more than 1 km above the stratocumulus deck, the area of positive radiative heating overlaps the area of extinction greater than $0.2\,\mathrm{km^{-1}}$ in BBRAD (Fig. 9b). Such strong heating increases the temperature which lowers the 316 K isentrope to 3 km altitude at 17° S.

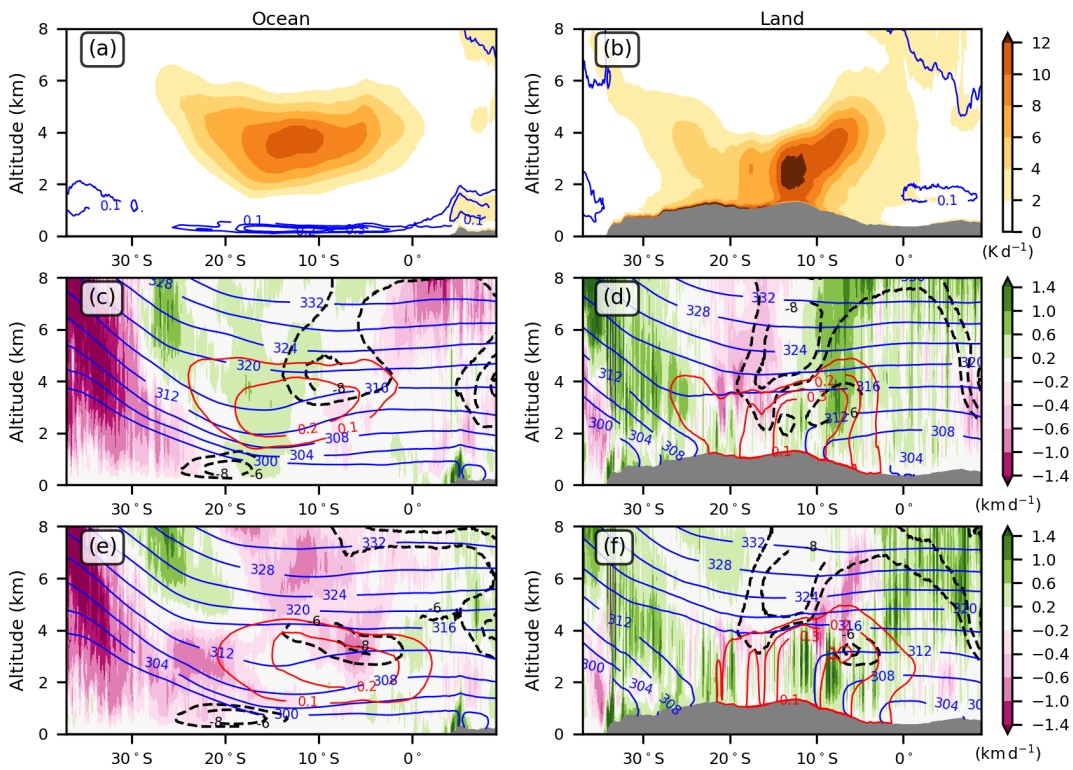

**Figure 9. (a–b)** Radiative heating (K d$^{-1}$, shading) and cloud fraction (blue contour) from BBRAD ensemble mean. **(c–f)** Vertical wind (m s$^{-1}$, shading), zonal wind (m s$^{-1}$, black contours when lower than $-6$ m s$^{-1}$, dotted when lower than $-10$ m s$^{-1}$), potential temperature (K, blue contours) and aerosol extinction (km$^{-1}$, red contours) from **(c–d)** BBRAD and **(e–f)** NORAD ensemble means. Fields are averaged at 12:00 UTC between 8 and 16 September 2017 and **(a, c, e)** between 5 and 10° E and **(b, d, f)** between 15 and 25° E.

This strengthens the meridional temperature gradient and accelerates the AEJ-S up to a core value of $-10$ m s$^{-1}$ at 10° S. The ageostrophic circulation associated with the AEJ-S induces upward motions to its south and downward motions to its north (Adebiyi and Zuidema, 2016). The self-lofting of BBA also reinforces the upward motions in the BBA layer. In NORAD, the extinction maximum of 0.2 km$^{-1}$ is 1 km lower and 5° northward than in BBRAD (Fig. 9c). The lack of additional BBA heating in NORAD results in a much smoother meridional temperature gradient. As a consequence, the zonal wind at mid-level is weaker and lower in altitude, while the Benguala low-level jet extends in a wider meridional band around 20° S. Between 10 and 20° S, the vertical motion is upward in BBRAD and downward in NORAD. This explains why the BBA reach a higher altitude in BBRAD.

A vertical perspective along longitude is shown across the stratocumulus deck and the BBA source areas between 5 and 15° S (Fig. 10). The vertical distribution is significantly modified by the radiative effect of the BBA. Over land, the largest extinction values are close to the ground in BBRAD while they extend to 4 km altitude in NORAD. Over the sea, they gradually rise to 4 km altitude in BBRAD and gradually fall to 2 km altitude in NORAD. The top of the cloud fraction increases westward in

BBRAD while remaining below 1 km altitude in NORAD. The separation between the westward extension of the AEJ-S and the eastward extension of the LLJ is at 0° longitude in BRAD and 10° E in BBRAD. Changes in the extinction distribution are significant almost everywhere below 6 km altitude (Fig. 10). The radiative effect of the BBAs leads to a significant temperature increase of 2 K between 2 and 4 km altitude and a decrease of 1 K below. The horizontal and vertical components of the wind speed are also significantly modified. The AEJ-S is accelerated, the LLJ is slowed down and the upward motions are

strengthened on land and around the coast. In the first km, both over land and sea, water vapor and cloud fraction increase, except along the coast where they decrease significantly. Over the sea, water vapor also increases between 2 and 4 km altitude. This shows the co-occurrence of smoke, temperature, wind and moisture anomalies.

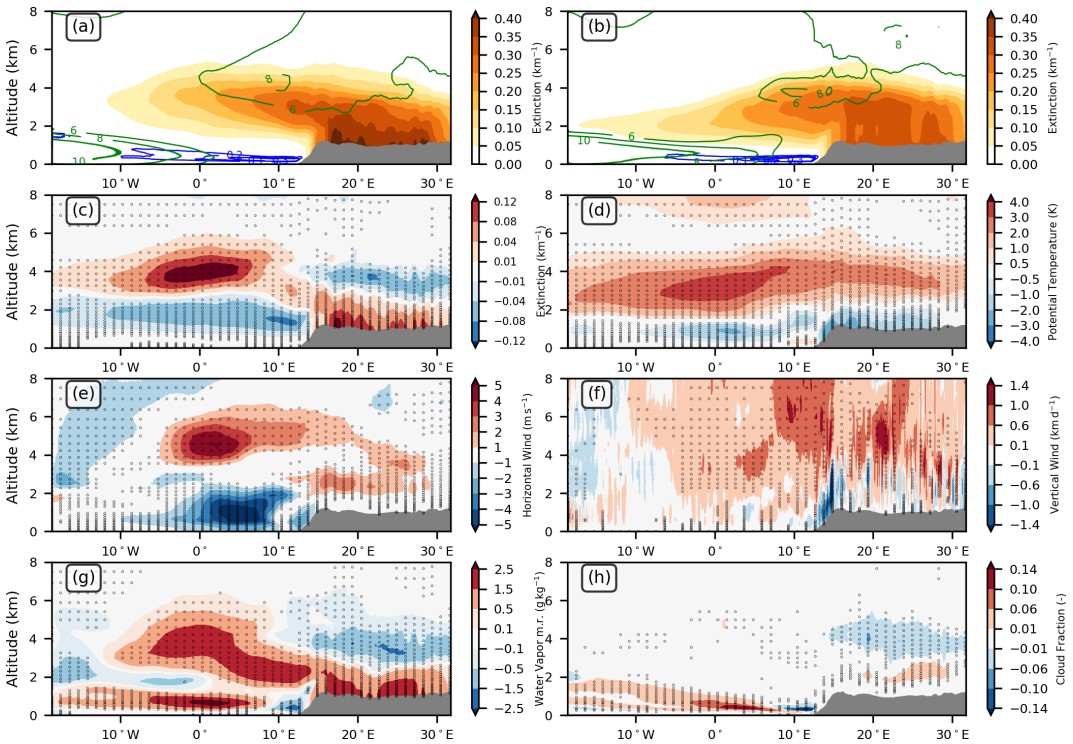

**Figure 10. (a–b)** Extinction (km$^{-1}$, shading), horizontal wind speed (m s$^{-1}$, green contour) and cloud fraction (blue contour) from **(a)** BBRAD **(b)** NORAD ensemble means. Changes between BBRAD and NORAD in **(c)** extinction, **(d)** potential temperature, **(e)** horizontal wind speed, **(f)** vertical wind, **(g)** water vapor mixing ratio and **(h)** cloud fraction. Black dots indicate where changes are statistically significant at the 0.05 level. Fields are averaged at 12:00 UTC between 8 and 16 September 2017 and between 5 and 15° S.

## 4.3 Impact on deep convection and rain

The radiative impact of BBA is shown for the temperature at 2 m, the wind at 10 m, convective available potential energy

(CAPE) and the rain rate (Fig. 11). Fields are daily mean between 8 and 16 September. The difference in 2 m temperature between the two ensembles is null over ocean because the sea surface temperature does not change over time for both simu-

lations (Fig. 11a). The overall change in circulation results in a (significant) cyclonic anomaly in the wind difference at 10 m, consistent with a limited northward extension of the Benguela low-level jet in BBRAD compared to NORAD. Over land, the 2 m temperature significantly decreases where the BBA dimming seen in Fig. 8a is the strongest. It reaches $-3$ K over Angola

and the southern part of the Democratic Republic of the Congo. This cooling effect has been reported in the literature (Mallet et al., 2020) as well as the cyclonic and anticyclonic anomalies in the wind difference at 10 m (see their Figure 11). The anticyclonic anomaly over land in Fig. 9a is not as well marked as the cyclonic circulation over the ocean, essentially due to surface friction. The 2 m temperature increases up to 1 to 2 K over land northward of 6° S, also significantly. This is an aerosol forcing that feeds back on deep convection, an aerosol effect already reported, but for dust over West Africa (Tompkins et al., 2005;

Chaboureau et al., 2007). Here, the anomaly prevents the low-level flow in BBRAD from extending as far northeast as it does in NORAD (Fig. 4f). This changes the location of the deep convective clouds by moving them to the northeast (Fig. 4a).

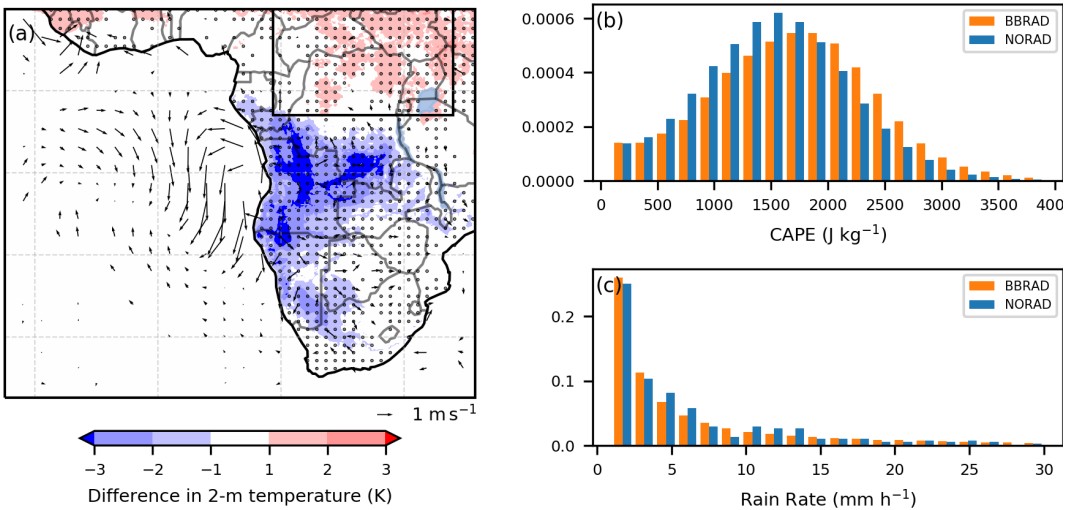

**Figure 11. (a)** Difference in 2 m-temperature (shading) and 10 m-wind (arrow) between BBRAD and NORAD ensemble means. Only statistically significant changes at the 0.05 level are shown for the 10 m-wind while the black dots over land overlay these changes for the 2 m temperature. Histogram of **(b)** CAPE and **(c)** rain rate. Fields are daily mean between 8 and 16 September 2017. In panels **(b)** and **(c)**, the fields are taken from the black box over land shown in panel **(a)**. In panel **(b)**, CAPE is shown for CIN less than 1 J kg$^{-1}$ and for CAPE larger than 100 J kg$^{-1}$. In panel **(c)**, the rain rate is shown for values between 1 and 30 mm h$^{-1}$.

Another effect of BBA on deep convection is its intensity as shown by the CAPE and rain rate between 14–36° E and 3° S– 10° N defining the black box shown in Fig. 11a. Because high temperatures are more frequent for BBRAD than for NORAD, the distribution in the most unstable CAPE varies with BBA (Fig. 11b). Following Reinares Martínez and Chaboureau (2018),

CAPE is shown for convective inhibition (CIN) less than 1 J kg$^{-1}$. Indeed, CAPE is accessible to an air parcel if it overcomes the CIN barrier. For a CIN greater than 1 J kg$^{-1}$, an air parcel would have a vertical wind speed greater than 1.4 m s$^{-1}$ below the level of free convection. In the lower levels of the atmosphere, updrafts generally do not exceed this vertical speed making the CAPE inaccessible to air parcels. As a result, around 21 % of the grid points have accessible CAPE for NORAD against

18 % for BBRAD. The larger number of grid points with accessible CAPE for NORAD explains the more frequent activity in deep convection in Western equatorial Africa (Fig. 4a). Since the 2 m temperature is higher in BBRAD than in NORAD, CAPE larger than 2000 J kg$^{-1}$ are more frequent in BBRAD. In other words, deep convection is most intense in BBRAD. However, medium values of CAPE around 1500 J kg$^{-1}$ are more numerous in NORAD resulting in more events with rain rate above 1 and 5 mm h$^{-1}$ (Fig. 11c). The more frequent rainfall events therefore explain why the 2 m temperature is lower in NORAD than in BBRAD in the black box shown in Fig. 11a.

## 4.4 Impact on transport

The radiative impact of BBA on transport is examined using the fraction of trajectories having undergone either convective or fire influences. This analysis is done over the last 7 d of the back-trajectories. Trajectories ascending by more than 3 km in 3 h at least once are considered to be under convective influence. This threshold, which is equivalent to the relatively small value of 0.3 m s$^{-1}$, is chosen to ensure that all convective updrafts are selected. Trajectories with BBA tracers are under fire influence if they pass over a fire region at less than 3 km altitude, i.e. within the continental boundary layer. Results are shown over the full domain and for four areas that include Ascension Island, São Tomé, St. Helena and Walvis Bay (Fig. 12). All areas have a size of 5°of longitude by 5°of latitude.

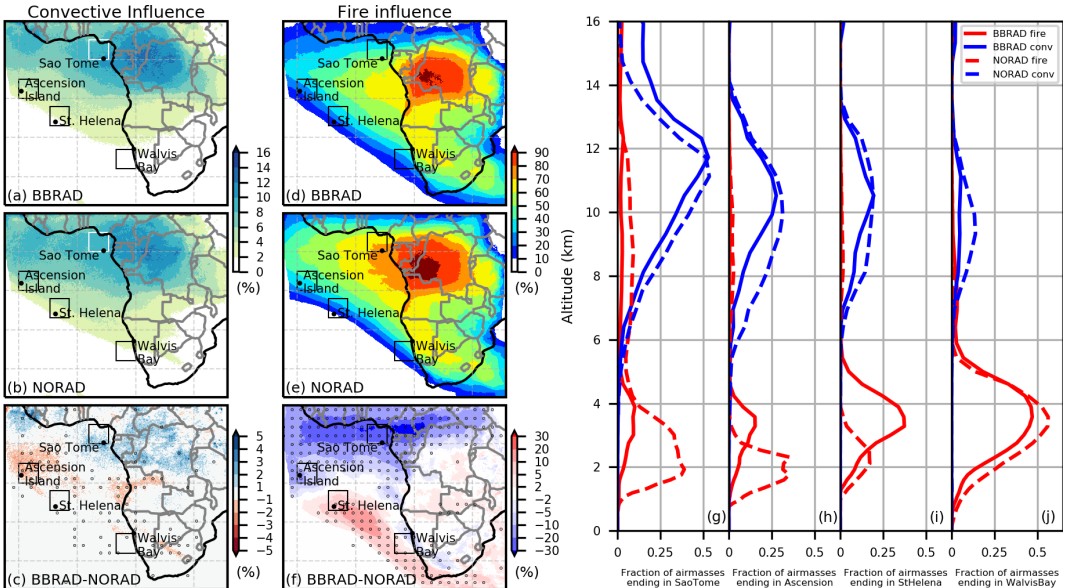

**Figure 12.** Left column: Fraction of air masses having **(a–b)** convective influence and **(d–e)** fire influence between 0 and 16 km in the last 7 d for **(a, d)** BBRAD, **(b, e)** NORAD and **(c, f)** their changes. In **(c, f)**, black dots indicate where changes are statistically significant at the 0.05 level. Right column: profiles of convective (blue) and fire (red) influences at **(g)** São Tomé, **(h)** Ascension, **(i)** St. Helena and **(j)** Walvis Bay areas for BBRAD (solid) and NORAD (dashed). Results are averaged from 7 d trajectories ending at 12:00 UTC every day between 8 and 16 September 2017.

The footprint of convective ascents is maximum over land in the northern part of the domain where deep convection occurs (Fig. 12a–b). The average convective influence between 0 and 16 km is greater than 16 % for both simulations. It extends over the Gulf of Guinea north of the equator within the tropical easterly jet. In the São Tomé area, the convective influence is limited to the altitude between 6 and 16 km in altitude (Fig. 12g). In consistency with more frequent but less intense deep convective activity, it peaks at 11 km altitude in NORAD instead of 12 km in BBRAD and is weaker above 11 km in NORAD. This explains the lower (but not significant) values of the vertical mean of the convective influence over equatorial Africa. Over the southeast Atlantic, deep convection has a remote influence on transport. It spreads over a larger area in NORAD, which extends downstream to Ascension Island and the eastern coasts of South Africa. Its maximum is more than 20 % at 11 km over Ascension Island and St. Helena, and 10 % at 10 km over Walvis Bay, but only in NORAD for the latter (Fig. 12h–j). It is worth noting that the convective influence extends to lower altitudes in NORAD, which is consistent with the higher subsidence over the southeast Atlantic found previously.

The fire influence covers the northern part of the southeast Atlantic and almost the entire African landmass with a maximum of 80 % over the Republic Democratic of the Congo (Fig. 12d–e). Over the Gulf of Guinea, it reaches a vertical average of 70 % in NORAD against 50 % in BBRAD. It extends further southwest in BBRAD with a peak at 50 % at about 4 km over St. Helena against 20 % at about 2 km in NORAD (Fig. 12i). This 20–30 % change in the horizontal distribution of fire influence is significant (Fig. 12f). It mimics that of the AOD, except for the amplitude. As an example, the fire influence at Walvis Bay is greater for NORAD than for BBRAD while the nearby AOD at Henties Bay is larger for BBRAD than for NORAD. Indeed, the fire influence does not account for the variability in BBA emission. Whatever the difference in amplitude, the effects due to the change in circulation remain the same: an export of BBA over the southeast Atlantic further west and at higher altitude when accounting for the radiative effect of BBA.

The pathways followed by the smoky air change if the radiative effect of BBA is taken into account. They are shown for the same four areas with median and interquartile values of the 7 d back trajectories ending between 8 and 16 September at 12:00 UTC (Fig. 13). Observing a single time of day highlights diurnal oscillations along the trajectories that peak during the day (Fig. 13b–e). This is due to the radiative effect experienced by the smoky air parcels. During the night, they cool down and descend. During the day, they heat up and ascend. Since the radiative heating is stronger for radiatively active BBA in BBRAD, the solar forcing causes a higher elevation and thus, a stronger diurnal oscillation.

Another effect of radiatively active BBA is that the trajectories of smoky air parcels ending at São Tomé, Ascension and St. Helena are higher in BBRAD than in NORAD (Fig. 13b–d). The average difference along the 7 d travel is 1190, 380, 250 and 260 m, respectively. This is consistent with the fire influence peaking at higher altitude at these locations. As a result of the stronger AEJ-S, their pathways have a more westward direction in BBRAD than in NORAD and have a more northerly origin (Fig. 13a). The stronger AEJ-S has a more dramatic impact on the trajectories ending at Walvis Bay (Fig. 13a). It transports BBA further west in BBRAD before being redirected to the southeast. As a result, the origin of BBA reaching the Walvis Bay area after 7 d of transport is Angola in BBRAD and Zambia in NORAD. The increased long-range transport of BBA is consistent with the faster and higher trajectories observed during strong jet events compared to weak jet events in the climatological study of Adebiyi and Zuidema (2016).

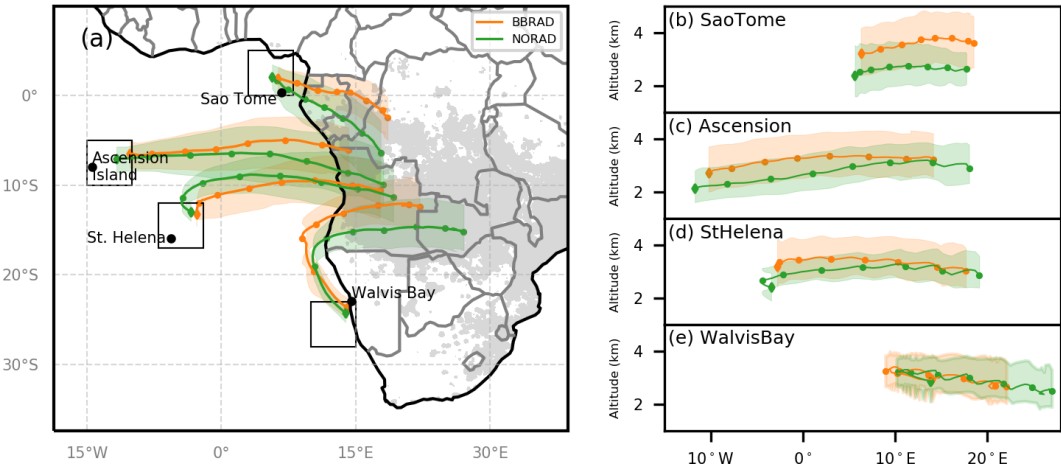

**Figure 13.** Back trajectories of smoky air parcels ending between 1 and 5 km in the São Tomé, Ascension, St. Helena and Walvis Bay areas. The median (bold lines, dots every 24 h, diamond at the end of the back trajectory) and the interquartile ranges (shading) are shown for the BBRAD (orange) and NORAD (green) ensembles. In panel **(a)**, fire sources are shaded in grey and interquartile ranges are for latitude. In panels **(b–e)**, the interquartile ranges are those of altitude. Results are averaged from 7 d trajectories ending at 12:00 UTC every day between 8 and 16 September 2017.

## 5 Conclusions

The direct and semi-direct effects of BBA over southern African and the southeast Atlantic is addressed during the AEROCLO-sA campaign in September 2017 with the use of two ensembles of convection-permitting simulations, one taking into account the BBA radiative effect and the other not. Their effects on the transport are also discussed using the Meso-NH online Lagrangian trajectory tool activated in both simulations.

The realism of both ensembles of simulations is assessed against satellite, airborne and ground-based observations and ECMWF analyzes. The BBRAD ensemble reproduces well the horizontal and vertical distribution of BBA as well as the temporal evolution of the AOD, at least for a 7 d period for the stations located far from the fire sources, namely São Tomé, Ascension Island and St. Helena. The location of the stratocumulus deck off Angola and the deep convective activity over equatorial Africa is also well captured. In contrast, the NORAD ensemble produces a BBA plume extending too far north, not far enough west and southwest and not high enough over the stratocumulus deck, a low-level cloud cover extending too far south and a deep convective activity more frequent and less intense than in BBRAD. The superiority of the simulations with BBA radiative effect and the strong significant effects induced by its deactivation allow a clear and high confidence assessment of their impact on dynamics and transport.

The direct radiative effect due to BBA is overall well simulated. At the surface, the dimming effect lowers the temperature by $-3$ K over Angola and the southern part of the Democratic Republic of Congo (while the sea surface temperature is prescribed in the simulations). At the top of the atmosphere, the direct radiative effect is strongest over the BBA layer covering a high

albedo surface, mainly the stratocumulus deck and the Congo forest. A daily average heating rate of more than $1\,\mathrm{K\,d^{-1}}$ is simulated in the continental boundary layer of the fire zone and in the BBA layer over the ocean at $4\,\mathrm{km}$ altitude.

   The first radiative impact of the BBA on the circulation is the acceleration of the AEJ-S. Due to the BBA radiative heating of the air above Angola and the Democratic Republic of the Congo, the meridional gradient of the temperature increases over land which in turn accelerates the AEJ-S due to thermal wind balance. The stronger AEJ-S moves the boundary between the

stratocumulus deck and the Benguala low-level jet further south. A second impact is the change in vertical circulation. The motion is upward over Angola and over the stratocumulus deck instead of downward when radiative effects are cut off. This can be explained by a combination of three causes: ageostrophic upward motion induced by the stronger AEJ-S; self-lofting of BBA; and reduced subsidence associated with less frequent deep convective activity over equatorial Africa.

   Another radiative impact of the BBA on circulation less studied in the literature is a modification of the BBA transport. As

shown with 7 d back-trajectories, it is found that their pathway is shifted by a few hundred kilometers, northward or westward depending on their final destination. This change considerably their areas of origin. Another result is that their transport altitude is a few hundred meters higher when the radiative effect of BBA is taken into account. This is because, by gaining heat by solar absorption during the day, they increase the potential temperature of the air in which they reside. In other words, BBA do not experience isentropic motions. Therefore, care must be taken when using Lagrangian trajectories along isentropes to study

origin and transport of BBA.

   Despite a rather coarse representation of BBA and their radiative effect in our simulations, our results suggest a significant impact on the dynamics, thermodynamics and composition of the atmosphere in southern Africa. Further studies with Meso-NH will investigate the sensitivity of the simulations to a more sophisticated representation of BBA and their optical and radiative properties, as well as their impact on regional dynamics and cloud distribution over land and ocean. The significance

of our results on the role of BBA radiative effects on the AEJ-S, will also be compared to longer climatological periods using climate simulations with and without BBA radiative forcing.

*Data availability.* The LNG lidar data is available via the digital object identifier (DOI): 10.6096/AEROCLO.1774 and the dropsondes data via DOI: 10.6096/AEROCLO.1777. The AERONET data were downloaded from the NASA AERONET website (http://aeronet.gsfc.nasa.gov/, last access: March 2022), the MODIS data from the Giovanni web portal (http://disc.sci.gsfc.nasa.gov/giovanni, last access: March 2022)

and the SEVIRI and CATS data from ICARE (https://www.icare.univ-lille.fr/, last access: March 2022). The Meso-NH-derived fields and back trajectories data can be obtained upon request to the corresponding author of the paper.

*Author contributions.* LL and JPC performed the simulations and the analyses. JPC prepared the manuscript with contributions from all co-authors.

*Competing interests.* The authors declare that they have no conflict of interest.

*Special issue statement.* This article is part of the special issue "New observations and related modelling studies of the aerosol–cloud–climate system in the Southeast Atlantic and southern Africa regions (ACP/AMT inter-journal SI)". It is not associated with a conference

*Acknowledgements.* Computer resources for running Meso-NH were allocated by GENCI through Project 90569. We thank the PIs Brent Holben, Jens Redemann, Carlos Ribeiro, Nichola Knox, Stuart Piketh, and their staff for establishing and maintaining the AERONET sites used in this investigation. JPC thanks NCAR/ACOM for the visitor grant. LL received a CNES postdoctoral fellowship. We thank the anonymous reviewers for their comments, which helped to improve the overall quality of the paper. The AEROCLO-sA project was supported by the French National Research Agency under grant agreement n° ANR-15-CE01-0014-01, the French national program LEFE/INSU, the Programme national de Télédetection Spatiale (PNTS, http://www.insu.cnrs.fr/pnts), grant n° PNTS-2016-14, the French National Agency for Space Studies (CNES), and the South African National Research Foundation (NRF) under grant UID 105958. The research leading to these results has received funding from the European Union's 7th Framework Programme (FP7/2014-2018) under EUFAR2 contract n°312609. Airborne data was obtained using the aircraft managed by Safire, the French facility for airborne research, an infrastructure of the French National Center for Scientific Research (CNRS), Météo-France and the French National Center for Space Studies (CNES). The strong diplomatic assistance of the French Embassy in Namibia, the administrative support of the Service Partnership and Valorisation of the Regional Delegation of the Paris–Villejuif region of the CNRS, and the cooperation of the Namibian National Commission on Research, Science and Technology (NCRST) were invaluable to make the project happen. The support of the aviation authorities is acknowledged. The AEROCLO-sA project would have not been successful without the endless efforts of all the research scientists and engineers involved in its preparation, often behind the scenes. Their support and enthusiasm are sincerely appreciated.

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

**Appendix A**

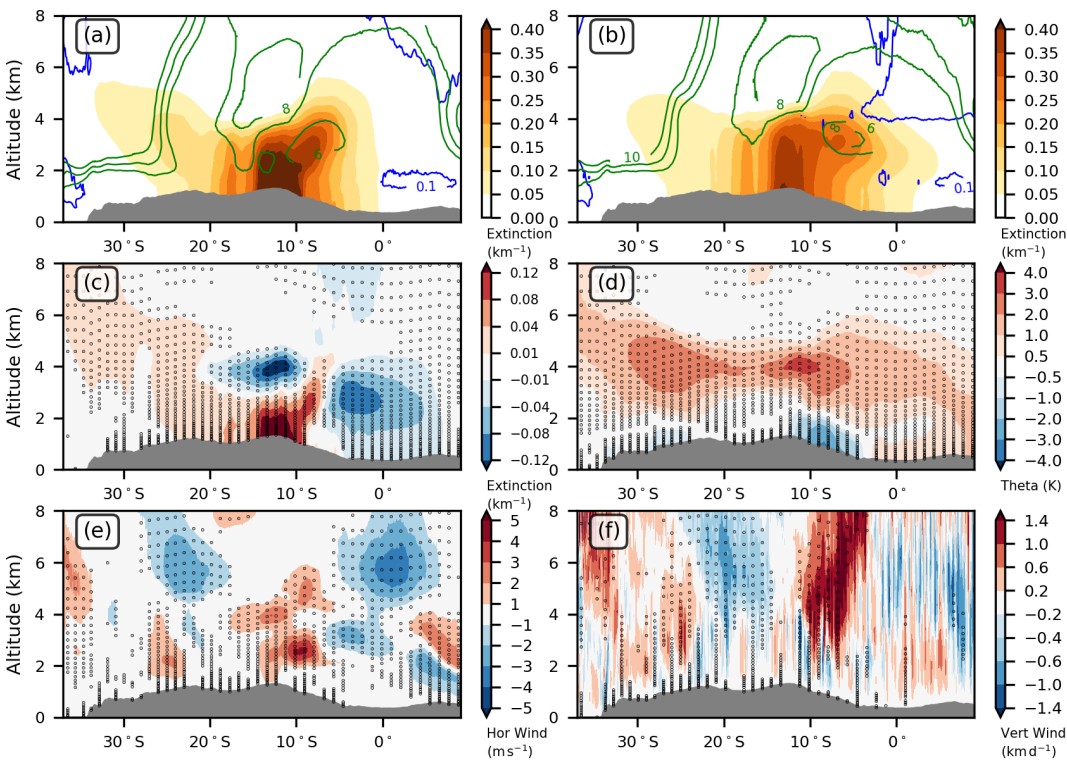

**Figure A1. (a–b)** Extinction (km$^{-1}$, shading), horizontal wind speed (m s$^{-1}$, green contour) and cloud fraction (blue contour) from **(a)** BBRAD **(b)** NORAD ensemble means. Changes between BBRAD and NORAD in **(c)** extinction, **(d)** potential temperature, **(e)** horizontal wind speed, and **(f)** vertical wind. Black dots indicate where changes are statistically significant at the 0.05 level. Fields are averaged at 12:00 UTC between 8 and 16 September 2017 and between 15 and 25° E.

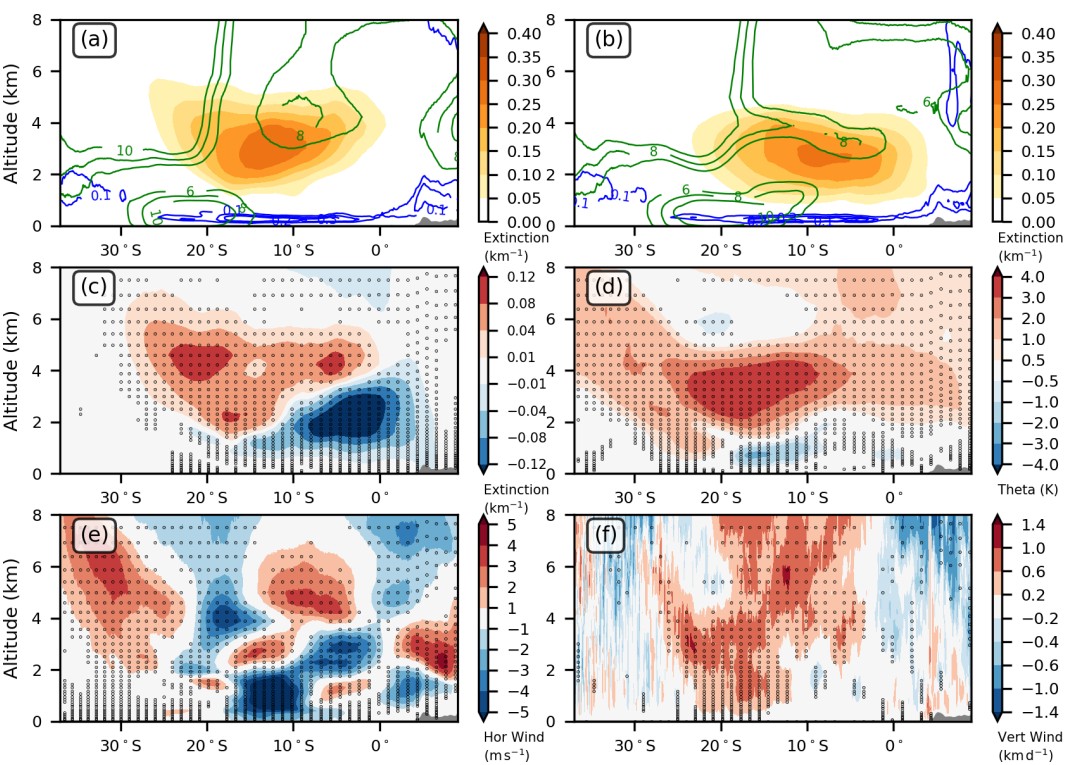

**Figure A2.** Same as Fig. A1 but for fields averaged between 5 and 10° E.