# Peer review of "Acceleration of the southern African easterly jet driven by radiative effect of biomass burning aerosols and its impact on transport during AEROCLO-sA"

_Atmospheric Chemistry and Physics, 2022_

## Referee Comment (RC1)

Summary comments:

In this manuscript the authors use two convection-permitting simulations one taking into account the BBA radiative effect and the other not to addressed the direct and semi-direct effects of BBA over southern African and the southeast Atlantic during AEROCLO-sA field campaign in September 2017. Both simulations are examined against satellite, airborne and ground-based observations and ECMWF analyses. Efforts are made to understand the acceleration of the southern components of the African Easterly Jet.
There is no doubt that such a comparison adds value to understanding overall dynamic of AEJ-S and heat low, which are linked to the direct radiative effect over Angola and Namibia. However, there are numerous (sometimes major) concerns with the analysis and approach. Therefore, I encourage the authors to significantly revise the paper.

Abstract:
The readability of the Abstract may be improved by using scientifically relevant terms such as "baroclinicity" and highlighting key mechanisms in general terms rather than discussing specific simulations issues.

Introduction :
The introduction is quite good and correctly motivates the study. But there are less relevant sentences and less citations. I strongly recommend looking at the introduction in a direct and succinct way to properly motivate the questions you hope to answer.

Data:
It is not clear why only September 2017 was used in the study. Please explain why.

Results section:
Dynamic and corresponding mathematical framework (e.g., Radiative heating, temperature gradient etc.) used in this study is unclear and makes it very difficult to understand the attributions/mechanisms for the differences in the AEJ-S intensity. Improvements are necessary here.

The authors need to clarify that method used to define the AEJ-S core to produce Figure 3. Also, clarification should be provided on why 8 m/s was used to identify AEJ-S.

Refine the colorbar in Figure. 1 so that all colors are used.

Clarify by explicitly writing the equation/formula used to create Figure5.

Section 3 was essentially descriptive and was correct. Section 4 deals with the direct effects of the BBA on the atmosphere - radiation distribution and circulation and have a better-developed process approach. I like your analyses. The main concern I have is about the perspective of coupling within the atmosphere (radiation, transport, rain and convection), in that the acceleration of the AEJ-S associated with radiative heating and increases the temperature through the intensity of the thermal heat low and gradient in temperature. So there must be some relationship and consistency among the section 3 and 4 analyses.

---

## Referee Comment (RC2)

Review of "Acceleration of the southern African easterly jet driven by radiative effect of biomass burning aerosols and its impact on transport during AEROCLO-sA" by Chaboureau et al.

This study uses a meso-scale model at convection-permitting scales to analyze the impact of aerosol-radiation interactions on simulated large-scale motions with a focus on African Easterly Jet-South (AEJ-S). The authors also use the same simulations to estimate the direct and semi-direct effects of aerosols over the southern African biomass burning and outflow region. Even though there is no lack of motivation for such a study and its usefulness for the modeling community in improving (or finding consensus) in estimating the impact of biomass burning aerosols (BBA) on radiation budget, clouds, and circulation/transport over this region. Especially since the vertical distribution of aerosols after long-range transport have been found to be lacking in many global models for this region.

However, there are serious concerns regarding the modeling approach and the use of over-simplistic assumptions regarding aerosol properties in the model, as mentioned below under my major comments. There is a need for some substantial modeling work to be redone/added before one can infer anything meaningful from the results and analysis presented in their Section 4 and beyond. I do not recommend the publication of the manuscript in its current form but can be reconsidered for review once additional modeling work is performed and analysis is redone.

**Major comments:**
When performing forecast runs or free-running simulations, it becomes necessary to perform ensemble runs (with larger number of ensemble members for shorter forecast periods) to understand the sensitivity of the model to the initial conditions in simulating/predicting the thermodynamic, wind and other meteorological fields. It is imperative to judge that how significant are the changes in wind speeds, temperature, cloud cover etc. (using a t-test for example) that the authors report between the two simulations versus the model internal variability.

Is the model assumption of SSA=0.85 at 532 nm? It would be worth demonstrating the assumed SSA curve for rest of the solar spectrum (or at least ~400-1000 nm, where absorption due to biomass burning and dust are most relevant). The model assumed SSA value is representative of strongly absorbing aerosols, possibly for the more aged smoke that was observed over the Ascension or during ORACLES. However, there is quite some variability of SSA over the continent (which is also part of the modeling domain), see Figure 11 of Piston et al. (2019). Therefore, I wonder if this strong heating (with SSA=0.85 everywhere) over the continent is exaggerating the changes in large-scale motions. The sensitivity of the model results to different SSA assumptions becomes critical here and should be discussed.

**Specific comments:**
**Lines 27-33:** There are several noteworthy references for modeling semi-direct effect over SE Atlantic in recent years, especially for the similar modeling scales authors are looking at. For

example, Lu et al. (2018), Gordon et al. (2018) and Das et al. (2020). These are worth citing in this section along with their major findings in relation to direct and semi-direct effect and how that compares to current study.

**Section 3.1, Figure 2:** Why is the spin-up time for aerosols considered within the 16d forecast period? A better or more common approach is to spin-up the aerosols for at least a 7–10-day period before the forecast, while the meteorology can be reinitialized using the ECMWF forecast from the starting point of forecast. This way the complete 16-day period can be used for analysis, which is what would make more sense to be able to make anything of the direct and semi-direct effect estimates presented later.

Further, not only AOD, but it would also be worth comparing the model SSA with retrievals from AERONET over these sites, including Mongu and Namibe sites. One would have assumed to include the two sites at the first place since they are the long-term sites over this region, with Mongu being a prime BB source example and Namibe has some considerable dust influence, which would be good to see provided the model includes dust as well. The resulting SSA (of the aerosol mixture, BBA + Dust) is important to constrain here since not only loading, but absorption/scattering will control the aerosol radiative impact and its feedback on the circulation.

**Fig 3.** Difference plots here would be useful for each row. Also, quantitative difference in windspeeds would be useful to demonstrate rather than qualitative description since the focus is on changes in AEJ-S based on the manuscript title/objectives.
Section 4.1. How do the DRE and radiative heating compare to other studies in literature? Authors just mention comparison to Mallet et al. currently. Also, 3-4 times higher values of DRE and radiative heating could also be because of higher model AOD magnitudes compared to MODIS and strongly absorbing SSA assumption. The seasonal mean versus September mean reasoning alone cannot explain such large differences.

**Fig. 8:** This figure is trying to present too much information in single panel (especially, c-d and e-f). Also, why meridional cross-sections are shown for this analysis when the AEJ-S flow is zonal? It would make more sense to show contours/cross-sections in the direction of plume transport (i.e., along longitude), starting from land and extending up to the ocean rather than meridional cross-sections over two narrow strips.

**Line 375 and 382:** The use of words like "first semi-direct effect" and "another radiative semi-direct effect" is misleading or inaccurate in these sentences. There is a whole lot of literature and history on how direct, semi-direct and indirect effect of aerosols have been defined. These effects have traditionally been defined with respect to the impact of aerosols on clouds and radiation balance. Unless authors are redefining what encompasses "semi-direct effect", they should refrain from using "semi-direct effect" arbitrarily or when implying the aerosol heating feedback on circulation rather.

Overall, it is hard to make anything of the changes discussed in 4.3 and beyond unless it is a comparison between model ensemble-means. It would benefit to show difference plots between your BBRAD and NORAD set of simulations for most of your variables (cloud fractions, wind speeds, temperature etc.) with significant changes depicted with hatching/shading. For example, change in surface temperature by 3K and average heating rates by 1K day$^{-1}$ (line 371) are too large and one would wonder if it's because of coarse representation of BBA in the model or due to the very short period of analysis with a single member run during which most fields have not stabilized in the model.

**Comment on Data availability:** The modeling results, at least the ones used for demonstrating the analysis presented within the paper should be uploaded and published using an appropriate data repository. The sentence stating, "The Meso-NH-derived fields and back trajectories data can be obtained upon request to the corresponding author of the paper" does not suffice based on current ACP guidelines on data policy.

**References:**

Das, S., Harshvardhan, H., and Colarco, P. R. (2020). The influence of elevated smoke layers on stratocumulus clouds over the SE Atlantic in the NASA Goddard Earth Observing System (GEOS) model. *Journal of Geophysical Research: Atmospheres*, 125, e2019JD031209. https://doi.org/10.1029/2019JD031209.

Gordon, H., Field, P. R., Abe, S. J., Dalvi, M., Grosvenor, D. P., Hill, A. A., Johnson, B. T., Miltenberger, A. K., Yoshioka, M., & Carslaw, K. S. (2018). Large simulated radiative effects of smoke in the south-east Atlantic. *Atmospheric Chemistry and Physics*, 18(20), 15,261–15,289. https://doi.org/10.5194/acp-18-15261-2018.

Lu, Z., Liu, X., Zhang, Z., Zhao, C., Meyer, K., Rajapakshe, C., Wu, C., Yang, Z., & Penner, J. E. (2018). Biomass smoke from southern Africa can significantly enhance the brightness of stratocumulus over the southeastern Atlantic Ocean. Proceedings of the National Academy of Sciences, 115(12), 2924–2929. https://doi.org/10.1073/pnas.1713703115

Pistone, K., Redemann, J., Doherty, S., Zuidema, P., Burton, S., Cairns, B., Cochrane, S., Ferrare, R., Flynn, C., Freitag, S., Howell, S. G., Kacenelenbogen, M., LeBlanc, S., Liu, X., Schmidt, K. S., Sedlacek III, A. J., Segal-Rozenhaimer, M., Shinozuka, Y., Stamnes, S., van Diedenhoven, B., Van Harten, G., and Xu, F. (2019). Intercomparison of biomass burning aerosol optical properties from in situ and remote-sensing instruments in ORACLES-2016, Atmos. Chem. Phys., 19, 9181–9208, https://doi.org/10.5194/acp-19-9181-2019 .

---

## Author Comment (AC1)

**Response to 'Comment on acp-2022-233' by Referee #1**

We thank the Referee for his/her time and his/her constructive comments. We have complied with most of the proposed changes. In the following, the comments made by the Referee appear in black, while our replies are in blue.

In this manuscript, the authors use two convection-permitting simulations one taking into account the BBA radiative effect and the other not to address the direct and semi-direct effects of BBA over southern Africa and the southeast Atlantic during the AEROCLO-sA field campaign in September 2017.

Both simulations are examined against satellite, airborne and ground-based observations and ECMWF analyses. Efforts are made to understand the acceleration of the southern components of the African Easterly Jet.

There is no doubt that such a comparison adds value to understanding the overall dynamic of AEJ-S and heat low, which are linked to the direct radiative effect over Angola and Namibia. However, there are numerous (sometimes major) concerns with the analysis and approach. Therefore, I encourage the authors to significantly revise the paper.

Abstract:

The readability of the Abstract may be improved by using scientifically relevant terms such as "baroclinicity" and highlighting key mechanisms in general terms rather than discussing specific simulations issues.
We revised the abstract and highlighted the radiative effect as the key mechanism of the study by rewording the two sentences as follows "The occurrence of stratocumulus over the southeast Atlantic, deep convective clouds over equatorial Africa and the large-scale circulation are all reproduced by the model. If the radiative effects of BBA are omitted in the model, we show that (i) the smoke plume is too low in altitude, (ii) the low-cloud cover is too weak, (iii) the deep convective activity is too frequent but not intense enough, (iv) the Benguela low-level jet is too strong, and (v) the southern African easterly jet is too weak. "

Introduction :

The introduction is quite good and correctly motivates the study. But there are less relevant sentences and less citations. I strongly recommend looking at the introduction in a direct and succinct way to properly motivate the questions you hope to answer.
The first sentence of the second paragraph was changed to read "BBA have also a semi-direct effect by affecting air temperature, atmospheric stability, low-level clouds and the regional atmospheric circulation." Following the suggestion of Referee #2, we added two citations: "Tummon et al. (2010) found a shallower boundary layer over the continent resulting from surface cooling combined with BBA-induced warming of the lower troposphere. [...] Das et al. (2020) found that elevating the BBA layer to higher levels, in agreement with lidar observations, increases oceanic cloudiness near the coast south of 10 S and decreases it far from the coast. [...] These changes in regional atmospheric circulation are crucial for the path of rivers of smoke, from the BBA sources in the tropics to their transport to the temperate mid latitudes and the southwestern Indian Ocean (Flamant et al. 2022)." To better explain our methodology, we added in the presentation of the outline of the paper: "It also evaluates the simulations against observations and shows the superiority of the simulation with radiatively-active BBA."

Data:

It is not clear why only September 2017 was used in the study. Please explain why.
We chose to focus our study on September 2017 because during this period we have the data from the AEROCLO-sA field campaign. This is explained in the penultimate paragraph of the introduction: "To achieve this objective, we investigate their effects using the airborne assets deployed during the AErosol, RadiatiOn, and CLOuds in southern Africa (AEROCLO-sA) field campaign (Formenti et al. 2019). From 5 to 12 September 2017, airborne lidar and dropsonde observations provided dedicated measurements of atmospheric dynamics, thermodynamics and aerosol composition."

Results section:

The dynamic and corresponding mathematical framework (e.g., Radiative heating, temperature gradient etc.) used in this study is unclear and makes it very difficult to understand the attributions/mechanisms for the differences in the AEJ-S intensity. Improvements are necessary here.

To quantify the effect of BBA on dynamics and physics, we added two new figures (Figs. 4 and 10 with the new numbering) that clearly show the changes in atmospheric dynamics, thermodynamics, and BBA loading due to the radiative effects of BBA. In addition, the significance of these changes is evaluated using two newly completed ensemble runs.

The authors need to clarify the method used to define the AEJ-S core to produce Figure 3. Also, clarification should be provided on why 8 m/s was used to identify AEJ-S.

Line 143 and following, introducing Fig. 3, we wrote "The fields are averaged between 8 and 16 September 2017 [...] The dynamics is represented by the wind field averaged at 12:00 UTC and taken at different altitudes: [...] 4 km is the altitude where the AEJ-S is maximum [...]". Line 165, we added: "The value of $-8\,\mathrm{m\,s^{-1}}$ is chosen because this threshold provides a good identification of the S-AEJ."

Refine the colour bar in Figure 1 so that all colours are used.

Figure 1 has been revised so that all colors are used to represent the carbon flux from biomass burning.

[Figure]

Figure 1 (revised): Meso-NH domain. The color shading shows the GFED emission of biomass burning carbon averaged between 1 and 16 September 2017.

Clarify by explicitly writing the equation/formula used to create Figure 5.

The left column of Fig. 5 shows the vertical cross-section of the extinction as explained in its legend. When we present Fig. 5, we now refer explicitly to the extinction. We describe the LNG extinction retrievals in Sec. 3.2 and refer to the Chazette et al. (2019) paper for the method. For the model, we write in Sec. 3.1 that we consider a mass extinction efficiency of $5.05\,\mathrm{m^2\,g^{-1}}$ for BBRAD, which multiplied by the BBA mass results in extinction.

Section 3 was essentially descriptive and was correct. Section 4 deals with the direct effects of the BBA on the atmosphere - radiation distribution and circulation and have a better-developed process approach. I like your analyses. The main concern I have is about the perspective of coupling within the atmosphere (radiation, transport, rain and convection), in that the acceleration of the AEJ-S is associated with radiative heating and increases the temperature through the intensity of the thermal heat low and gradient in temperature. So there must be some relationship and consistency among the sections 3 and 4 analyses.

To clearly distinguish the content of Secs. 3 and 4, we added in the last paragraph of the introduction: "Section 3 gives an overview of aerosols, clouds and dynamics during the 16 d period. It also evaluates the simulations against observations and shows the superiority of the simulation with radiatively-active BBA." More references to Sec. 4 were added in Sec. 3 to better link the two sections.

---

## Author Comment (AC2)

**Response to 'Comment on acp-2022-233' by Referee #2**

We thank the Referee for his/her time and his/her constructive comments. We have complied with most of the proposed changes. In the following, the comments made by the Referee appear in black, while our replies are in blue.

This study uses a meso-scale model at convection-permitting scales to analyze the impact of aerosol-radiation interactions on simulated large-scale motions with a focus on African Easterly Jet-South (AEJ-S). The authors also use the same simulations to estimate the direct and semi-direct effects of aerosols over the southern African biomass burning and outflow region. Even though there is no lack of motivation for such a study and its usefulness for the modeling community in improving (or finding consensus) in estimating the impact of biomass burning aerosols (BBA) on radiation budget, clouds, and circulation/transport over this region. Especially since the vertical distribution of aerosols after long-range transport have been found to be lacking in many global models for this region.

We thank the Referee for recognizing the interest of our study and its focus on the Southern African Easterly Jet. Because of his/her interest in the vertical distribution of aerosols, we added a new figure (Fig. 10 with the new numbering) to discuss the radiative effects of BBA on their vertical distribution.

However, there are serious concerns regarding the modeling approach and the use of over-simplistic assumptions regarding aerosol properties in the model, as mentioned below under my major comments. There is a need for some substantial modeling work to be redone/added before one can infer anything meaningful from the results and analysis presented in their Section 4 and beyond. I do not recommend the publication of the manuscript in its current form but can be reconsidered for review once additional modeling work is performed and analysis is redone.

We agree with the Referee that we have used a somewhat simplified representation of BBA aerosol properties in order to perform high-resolution cloud-resolving simulations. However, our assumptions about aerosol properties are quite realistic, as shown by many comparisons with observations. It is common to use simplistic parameterizations of, for example, convection or aerosol chemistry in chemistry-climate models when studying aerosol radiative effects. We have performed additional analysis to address the Referee's concerns, and we hope that our responses below will convince the Referee of the robustness of our results and of the significance of the radiative effect of BBA in accelerating the S-AEJ and changing the regional circulation over southern Africa and the southeast Atlantic.

**Major comments:**

When performing forecast runs or free-running simulations, it becomes necessary to perform ensemble runs (with larger number of ensemble members for shorter forecast periods) to understand the sensitivity of the model to the initial conditions in simulating/predicting the thermodynamic, wind and other meteorological fields. It is imperative to judge that how significant are the changes in wind speeds, temperature, cloud cover etc. (using a t-test for example) that the authors report between the two simulations versus the model internal variability.

We agree that the significance of any change can be estimated using ensemble averages. Therefore, we performed sensitivity simulations to initial conditions following the methodology of Das et al. (2020) and included the associated results (Figs. 4 and 10 with the new numbering and two other figures in Appendix A) in the revised paper. This allows us to test the significance of changes due to BBA radiative effects, as requested.

Is the model assumption of SSA=0.85 at 532 nm? It would be worth demonstrating the assumed SSA curve for rest of the solar spectrum (or at least 400-1000 nm, where absorption due to biomass burning and dust are most relevant). The model assumed SSA value is representative of strongly absorbing aerosols, possibly for the more aged smoke that was observed over the Ascension or during ORACLES. However, there is quite some variability of SSA over the continent (which is also part of the modeling domain), see Figure 11 of Piston et al. (2019). Therefore, I wonder if this strong heating (with SSA=0.85 everywhere) over the continent is exaggerating the changes in large-scale motions. The sensitivity of the model results to different SSA assumptions becomes critical here and should be discussed.

The SSA value of 0.85 assumed by the model is indeed representative of strongly absorbing aerosols. In Sec. 2.1

(lines 82–84), we wrote "This value, which corresponds to strongly absorbing aerosols, is close to the vertical average estimated at Ascension Island (Wu et al. 2020) and over the southeastern Atlantic (Pistone et al. 2019, Cochrane et al. 2022)." Following your suggestion regarding "Section 3.1, Figure 2", we revised Fig. 2 (see below) by including the AERONET and BBRAD SSAs. We added "The SSA is also shown for AERONET and BBRAD with dotted lines. In order to produce a column SSA value for comparison with AERONET, the BBRAD simulated SSA is averaged after weighting its value according to the aerosol optical depth at each vertical level." Figure 2 shows that the SSA varies with station and the BBRAD simulation captures the observed variation of SSA with station well.

**Specific comments:**

**Lines 27-33:** There are several noteworthy references for modeling semi-direct effect over SE Atlantic in recent years, especially for the similar modeling scales authors are looking at. For example, Lu et al. (2018), Gordon et al. (2018) and Das et al. (2020). These are worth citing in this section along with their major findings in relation to direct and semi-direct effect and how that compares to current study.
We thank the Referee for the references. We added three new references (Das et al. 2020; Tummon et al. 2010; Key and Haywood 2003). The studies of Lu et al. (2018) and Gordon et al. (2018) address the aerosol-cloud interaction that we do not consider (line 54).

**Section 3.1, Figure 2:** Why is the spin-up time for aerosols considered within the 16d forecast period? A better or more common approach is to spin-up the aerosols for at least a 7–10-day period before the forecast, while the meteorology can be reinitialized using the ECMWF forecast from the starting point of forecast. This way the complete 16-day period can be used for analysis, which is what would make more sense to be able to make anything of the direct and semi-direct effect estimates presented later.
The spin-up time for aerosols varies with location, as shown by comparison with AERONET AOD (Fig. 2). It is only 1 d for stations over or near fire source areas (e.g., Sakeji and Lubango) and 7 d for remote stations (e.g., Ascension Island and St. Helena). This 7 d spin-up time explains why we show fields averaged between 8 and 16 September. This also allows the BBAs to have a full radiative impact on the atmospheric circulation. Starting the numerical experiments with the same meteorological conditions, but with a different BBA loading, would reduce the overall effect on atmospheric conditions. Instead, the approach used here maintains consistency between the BBAs and the meteorological fields, which is essential as the BBAs feedback on the circulation.
In Sec. 2.1, we added "The 7 first day is used to spin the model up and the results are shown for averages between 8 and 16 September. It corresponds to the time required for the BBAs to reach the westernmost islands in the Atlantic, as shown below. It also allows the BBAs to have a full radiative impact on the atmospheric circulation while keeping them consistent with temperature and the winds."

Further, not only AOD, but it would also be worth comparing the model SSA with retrievals from AERONET over these sites, including Mongu and Namibe sites. One would have assumed to include the two sites at the first place since they are the long-term sites over this region, with Mongu being a prime BB source example and Namibe has some considerable dust influence, which would be good to see provided the model includes dust as well. The resulting SSA (of the aerosol mixture, BBA + Dust) is important to constrain here since not only loading, but absorption/scattering will control the aerosol radiative impact and its feedback on the circulation.
Following your suggestion, Fig. 2 was revised (see below). First, the selection of AERONET sites includes Namibe and Mongu (at the expense of Bamenda and Henties Bays). Second, the SSA is shown in addition to AOD. Over Mongu, the BBRAD value (0.87) is greater than that of AERONET (0.85). Over Namibe, no dust is simulated and the SSA value is observed and simulated around 0.86: this shows the absence of a "considerable dust influence" for the period considered in our study. Overall, BBRAD reproduces the AERONET AOD and SSA values for most stations. This makes us confident about the optical properties of BBA that we use.

**Fig 3.** Difference plots here would be useful for each row. Also, quantitative difference in windspeeds would be useful to demonstrate rather than qualitative description since the focus is on changes in AEJ-S based on the manuscript title/objectives.
We added four new figures (Figs. 4 and 10 with the new numbering and two other figures in Appendix A) showing the differences and their significance.

**Section 4.1.** How do the DRE and radiative heating compare to other studies in literature? Authors just mention comparison to Mallet et al. currently. Also, 3-4 times higher values of DRE and radiative heating could also be because of higher model AOD magnitudes compared to MODIS and strongly absorbing SSA assumption. The seasonal mean versus September mean reasoning alone cannot explain such large differences.

We now refer to Sakaeda et al. (2011), in addition to Mallet et al. (2020), for the seasonally averaged DRE values of $-30\,\mathrm{W\,m^{-2}}$. Note that we incorrectly calculated the ratio of the September value of $-100\,\mathrm{W\,m^{-2}}$ to the seasonal value of $-30\,\mathrm{W\,m^{-2}}$. This ratio is 2–3 times, and not 3–4 times. We agree that the high values of AOD can explain the large values of DRE. Lines 243–244, we wrote "The larger magnitude is explained by both the greater insolation and AOD in mid-September compared to seasonal values." Also note that the AOD comparison with AERONET does not show any high model BBA values, nor does the newly added SSA comparison with AERONET. We added "In terms of radiative forcing efficiency, i.e. DRE divided by AOD, it is between $-10$ and $-20\,\mathrm{W\,m^{-2}\,\tau^{-1}}$, a value consistent with the estimate of Sakaeda et al. (2011)"

**Fig. 8:** This figure is trying to present too much information in single panel (especially, c-d and e-f). Also, why meridional cross-sections are shown for this analysis when the AEJ-S flow is zonal? It would make more sense to show contours/cross-sections in the direction of plume transport (i.e., along longitude), starting from land and extending up to the ocean rather than meridional cross-sections over two narrow strips.

Figure 8 is intended to show the key, albeit numerous, variables explaining the main changes between the two simulations. It is a meridional cross-section because it allows for discussion of the S-AEJ, a zonal thermal wind. When introducing Fig. 8, we added "The latitude-height cross-section is used to show the S-AEJ thermal wind for which the zonal wind varies with the meridional temperature gradient." The directions of the plume transport are shown in Fig. 11 using trajectories. Some are effectively westward, others turn southward.

**Line 375 and 382:** The use of words like "first semi-direct effect" and "another radiative semi-direct effect" is misleading or inaccurate in these sentences. There is a whole lot of literature and history on how direct, semi-direct and indirect effect of aerosols have been defined. These effects have traditionally been defined with respect to the impact of aerosols on clouds and radiation balance. Unless authors are redefining what encompasses "semi-direct effect", they should refrain from using "semi-direct effect" arbitrarily or when implying the aerosol heating feedback on circulation rather.

The wording "semi-direct effect" was changed to "BBA impact on circulation".

Overall, it is hard to make anything of the changes discussed in 4.3 and beyond unless it is comparison between model ensemble-means. It would benefit to show difference plots between your BBRAD and NORAD set of simulations for most of your variables (cloud fractions, wind speeds, temperature etc.) with significant changes depicted with hatching/shading. For example, change in surface temperature by 3K and average heating rates by 1K day-1 (line 371) are too large and one would wonder if it's because of coarse representation of BBA in the model or due to the very short period of analysis with a single member run during which most fields have not stabilized in the model.

We disagree that the daily heating rate is too high. As written line 255, "Overall, the heating rates due to BBA are within the range of daily values around $1\,\mathrm{K\,day^{-1}}$ reported by Mallet et al. (2020)." (This is the value that is repeated line 371 in the conclusion). This value was also obtained by Tummon et al 2010 (see their figure 7). We added this reference in the text as well the reference to Keil and Haywood (2003), who found an even larger heating rate of $1.77\,\mathrm{K\,day^{-1}}$.

We disagree that the surface temperature is too high. As written line 294, "the cooling effect of $-3$ K has been reported in the literature (Mallet et al. 2020)."

The BBAs are indeed represented in a coarse way using a mass tracer. However, their optical properties are correct in the BBRAD ensemble, as shown by the comparison with AERONET, and their vertical distributions are correct, as shown by the comparison with lidar observations.

The dynamics and thermodynamics fields are "stable" as much as a chaotic atmosphere can be. There is no indication of "not stabilized fields" in the average fields of either. Following your suggestion, we calculated ensemble-means. As intuitively expected, the changes are significant where the differences in temperature, wind, etc. are large. This is shown for some additional figures. Note that the period is in September, when the S-AEJ maximum coincides with the AOD maximum over southern Africa (line 37). This largely explains why the circulation shows a strong response to the radiative effects of the BBA.

**Comment on Data availability:** The modeling results, at least the ones used for demonstrating the analysis presented within the paper should be uploaded and published using an appropriate data repository. The sentence stating, "The Meso-NH-derived fields and back trajectories data can be obtained upon request to the corresponding author of the paper" does not suffice based on current ACP guidelines on data policy.

This sentence echoes that recently published in the ACP paper by Flamant et al. (2022). It is planned to upload a subset of the simulations to the AERIS data center which hosts the AEROCLO-sA database. A data paper is currently being written.

**References:**

Das, S., Harshvardhan, H., and Colarco, P. R. (2020). The influence of elevated smoke layers on stratocumulus clouds over the SE Atlantic in the NASA Goddard Earth Observing System (GEOS) model. Journal of Geophysical Research: Atmospheres, 125, e2019JD031209. https://doi.org/10.1029/2019JD031209.

Gordon, H., Field, P. R., Abe, S. J., Dalvi, M., Grosvenor, D. P., Hill, A. A., Johnson, B. T., Miltenberger, A. K., Yoshioka, M., and Carslaw, K. S. (2018). Large simulated radiative effects of smoke in the south-east Atlantic. Atmospheric Chemistry and Physics, 18(20), 15,261–15,289. https://doi.org/10.5194/acp-18-15261-2018.

Lu, Z., Liu, X., Zhang, Z., Zhao, C., Meyer, K., Rajapakshe, C., Wu, C., Yang, Z., and Penner, J. E. (2018). Biomass smoke from southern Africa can significantly enhance the brightness of stratocumulus over the southeastern Atlantic Ocean. Proceedings of the National Academy of Sciences, 115(12), 2924–2929. https://doi.org/10.1073/pnas.1713703115

Pistone, K., Redemann, J., Doherty, S., Zuidema, P., Burton, S., Cairns, B., Cochrane, S., Ferrare, R., Flynn, C., Freitag, S., Howell, S. G., Kacenelenbogen, M., LeBlanc, S., Liu, X., Schmidt, K. S., Sedlacek III, A. J., Segal-Rozenhaimer, M., Shinozuka, Y., Stamnes, S., van Diedenhoven, B., Van Harten, G., and Xu, F. (2019). Intercomparison of biomass burning aerosol optical properties from in situ and remote-sensing instruments in ORACLES-2016, Atmos. Chem. Phys., 19, 9181–9208, https://doi.org/10.5194/acp-19-9181-2019 .

Flamant, C., M. Gaetani, J.-P. Chaboureau, P. Chazette, J. Cuesta, S. J. Piketh, and P. Formenti, Smoke in the river: an Aerosols, Radiation and Clouds in southern Africa (AEROCLO-sA) case study, Atmos. Chem. Phys., 22, 5701–5724, 2022, doi:10.5194/acp-22-5701-2022

Keil, A., and Haywood, J. M. (2003), Solar radiative forcing by biomass burning aerosol particles during SAFARI 2000: A case study based on measured aerosol and cloud properties, J. Geophys. Res., 108, 8467, doi:10.1029/2002JD002315, D13.

Mallet, M., Solmon, F., Nabat, P., Elguindi, N., Waquet, F., Bouniol, D., Sayer, A. M., Meyer, K., Roehrig, R., Michou, M., Zuidema, P., Flamant, C., Redemann, J., and Formenti, P.: Direct and semi-direct radiative forcing of biomass-burning aerosols over the southeast Atlantic (SEA) and its sensitivity to absorbing properties: a regional climate modeling study, Atmos. Chem. Phys., 20, 13 191–13 216, https://doi.org/10.5194/acp-20-13191-2020, 2020.

Sakaeda, N., Wood, R., and Rasch, P. J.: Direct and semidirect aerosol effects of southern African biomass burning aerosol, J. Geophys. Res., 116, https://doi.org/10.1029/2010JD015540, 2011.

Tummon, F., Solmon, F., Liousse, C., and Tadross, M.: Simulation of the direct and semidirect aerosol effects on the southern Africa regional climate during the biomass burning season, J. Geophys. Res., 115, D19206, https://doi.org/10.1029/2009JD013738, 2010.

[Figure]

Figure 2 (revised): Time evolution of daily mean AOD at 532 nm between 1 and 16 September 2017 from AERONET (blue), BBRAD (orange) and NORAD (green) at **(a)** Ascension Island, **(b)** Sakeji, **(c)** St. Helena, **(d)** Mongu, **(e)** São Tomé, **(f)** Lubango, **(g)** Namibe, **(h)** Windpoort, **(i)** Gobabeb and **(j)** Tsumkwe. The orange and green thin lines show the AOD due to dust for BBRAD and NORAD, respectively. The blue and orange dotted lines show the SSA at 440 nm for AERONET and BBRAD, respectively. Results are shown for the BBRAD and NORAD members starting at 00:00 UTC 01 September 2017.

---

## Author Comment (AC3)

**Response to 'Comment on acp-2022-233' by Editor Paquita Zuidema**

We thank the Editor for her time and her constructive comments. We have complied with most of the proposed changes. In the following, the comments made by the Editor appear in black, while our replies are in blue.

Both reviewers recommended major revisions, with the 2nd referee suggesting an additional review. As such I am asking the same referees to reevaluate the manuscript. I personally think you have done a good job addressing the referee concerns. The writing is more polished, and the additional work of adding in additional simulations to generate ensembles is valuable and appreciated. I do have some small additional comments of my own on the revised version, listed below.

P. 2 line 35: 'an increase' in what? Changed. The sentence is now: "Hodzic and Duvel (2018) found reduced deep convection over a tropical island and convergence of water vapor toward the island for moderately absorbing BBAs and increased deep convection for more strongly absorbing BBAs."

p.2 line 39: the changes in regional circulation also affects aerosol transport over the SEA of course....e.g. the ability to reach south America (Holanda et al., 2020, ACP https://acp.copernicus.org/articles/20/4757/2020/). Added

fig. 4 e and f: wind vectors difficult to read. We improved the readability of wind vectors.

p. 12 line 258: remove 'there'. Done

P. 15 line 300: What is the night-time cooling effect? Smoke doesn't have a long wave signature. Is this from water vapor? Is an altered water vapor transport also a feature of the AEJ-S in these simulations? It is indeed a nighttime cooling effect in the LW mostly due to water vapor. The changes in water vapor mixing ratio and cloud fraction along longitude are now shown in Fig. 10.

p. 15 line 309: an → a. Changed

P. 19 line 386: 'In consistency with a' → 'consistent with'. Changed

the authors may also want to consider how this work relates to Kuete et al., 2021 https://link.springer.com/article/10.1007/s00382-019-05072-w. In addition Ryoo et al 2021 https://acp.copernicus.org/articles/21/16689/2021/ provides some climatological context for the focus on September 2017 (mainly shows September 2017 had a slightly weaker AEJ-S than the climatological mean.) should that be of interest. We added these two references in the introduction.

---

## Author Response (AR2)

**Response to 'Comment on the revised version acp-2022-233' by Referee #2**

We thank the Referee for his/her time and his/her constructive comments. We have complied with most of the proposed changes. In the following, the comments made by the Referee appear in black, while our replies are in blue.

Review of the revised manuscript of "Acceleration of the southern African easterly jet driven by radiative effect of biomass burning aerosols and its impact on transport during AEROCLO-sA" by Chaboureau et al.

The authors have made major changes in their revised manuscript, especially by adding ensemble members and thereby providing the needed confidence in the presented results and analysis. They also addressed most of my previous comments satisfactorily in their revisions and review responses. The literature review part is also improved, and the writing is overall clearer and more coherent than it was before. I recommend the publication of the revised manuscript in ACP.

I only have minor/editorial comments below that should be considered before publication:

Figure 6: The changes in extinction between the BBRAD and NORAD are explained using the back-trajectory analysis of the air masses observed along LNG track. I wonder if there is also a role of associated water vapor transport that is causing the differences in extinction? For example, are the AOD and extinction differences between BBRAD and NORAD due to more smoke mass being transported in BBRAD at these levels or the humidity is also higher in BBRAD plume than NORAD? It would be interesting to add a panel here or elsewhere, depicting the changes in specific humidity/ RH between BBRAD and NORAD along AEJ-S. Changes of water vapor mixing ratio and cloud fraction along longitude are now shown in Fig. 10. In the text, we added "In the first km, both over land and sea, water vapor and cloud fraction increase, except along the coast where they decrease significantly. Over the sea, water vapor also increases between 2 and 4 km altitude. This shows the co-occurrence of smoke, temperature, wind and moisture anomalies." (In the simulations, BB extinction does not vary with relative humidity.)

Line 247: "Note that no dust is simulated along the leg for both simulations." This sentence is confusing. Did you mean that "no dust" is observed in the simulations even though the track is off of Namibe or did you not include the dust component for these simulations? In that case, is it a different set of simulations than presented elsewhere? Changed to "no dust is found along the leg for both simulations"

Line 352: "Another semi-direct effect of BBA on deep convection". Replace/omit "semi-direct". Deleted

Figure 12: The figure caption does not refer to the correct figure panels. It would also be useful to add the name of the location (Ascension Island, São Tomé, St. Helena, and Walvis Bay) on each profile panel. Changed

Line 422: "globally" → overall. It is not a global simulation. Changed